# Phase transitioned nuclear Oskar promotes cell division of *Drosophila* primordial germ cells

Kathryn E Kistler[1,2†], Tatjana Trcek[1†*], Thomas R Hurd[1,3], Ruoyu Chen[1], Feng-Xia Liang[4,5], Joseph Sall[5], Masato Kato[6], Ruth Lehmann[1,4*]

[1]Skirball Institute of Biomolecular Medicine, Howard Hughes Medical Institute, NYU School of Medicine, New York, United States; [2]Department of Molecular and Cellular Biology, University of Washington, Washington, United States; [3]Department of Molecular Genetics, University of Toronto, Toronto, Canada; [4]Department of Cell Biology, NYU School of Medicine, New York, United States; [5]DART Microscopy Laboratory, NYU Langone Health, New York, United States; [6]Department of Biochemistry, University of Texas Southwestern Medical Center, Texas, United States

*For correspondence:
Tatjana.TrcekPulisic@med.nyu.edu (TT);
lehmann@saturn.med.nyu.edu (RL)

†These authors contributed equally to this work

Competing interests: The authors declare that no competing interests exist.

**Abstract** Germ granules are non-membranous ribonucleoprotein granules deemed the hubs for post-transcriptional gene regulation and functionally linked to germ cell fate across species. Little is known about the physical properties of germ granules and how these relate to germ cell function. Here we study two types of germ granules in the *Drosophila* embryo: cytoplasmic germ granules that instruct primordial germ cells (PGCs) formation and nuclear germ granules within early PGCs with unknown function. We show that cytoplasmic and nuclear germ granules are phase transitioned condensates nucleated by Oskar protein that display liquid as well as hydrogel-like properties. Focusing on nuclear granules, we find that Oskar drives their formation in heterologous cell systems. Multiple, independent Oskar protein domains synergize to promote granule phase separation. Deletion of Oskar's nuclear localization sequence specifically ablates nuclear granules in cell systems. In the embryo, nuclear germ granules promote germ cell divisions thereby increasing PGC number for the next generation.
DOI: https://doi.org/10.7554/eLife.37949.001

## Introduction

Specialized ribonucleoprotein (RNP) granules are a hallmark of all germ cells. Throughout the animal kingdom, these granules share germline-specific proteins such as the ATP-dependent RNA helicase Vasa and are present during different stages of germ cell development. Morphologically, germ granules resemble other non-membrane bound RNP condensates such as P bodies and stress granules (*Courchaine et al., 2016*; *Decker and Parker, 2012*). Biophysical studies describe membrane-less granules as droplets that form by liquid-liquid phase separation (LLPS) where proteins phase separate by concentration-dependent de-mixing thereby markedly increasing their concentration within the condensate compared to the solution (*Brangwynne, 2013*; *Brangwynne et al., 2009*; *Courchaine et al., 2016*; *Hyman et al., 2014*). Proteins engaged in these transitions exchange freely with their environment. Transitions to a more structured, highly polymerized phase have been described for FUS (Fused in sarcoma) granules found in ALS patients (*Han et al., 2012*; *Kato et al., 2012*; *Murray et al., 2017*), Rim4 amyloids in meiotic fission yeast (*Berchowitz et al., 2015*) and the FG repeat proteins of the nuclear pore complex (*Frey et al., 2006*). Here concentration-dependent phase transitions lead from a liquid to a less fluid hydrogel-like state, where at least

a fraction of granule protein is polymerized into β-sheet filaments (*Kato et al., 2012*; *Lin et al., 2016*). Because of this, the proteins that reside within hydrogels are more stably associated and only slowly exchange with their environment (*Kato et al., 2012*). Proteins that mediate these concentration dependent transitions often contain low complexity (LC) domains (domains that contain an over-representation of a subset of amino acids (aa) in the primary protein sequence) or intrinsically disordered regions (IDRs), which promote non-specific protein-protein interactions, as well as RNA recognition motifs (RRMs), posttranslational modifications and higher specificity dimerization and protein-protein interaction domains (*Banani et al., 2016*; *Brangwynne, 2013*; *Courchaine et al., 2016*). P granules, the embryonic germ granules of *C. elegans,* are composed of different LC and IDR domain containing proteins and behave largely as condensed liquid droplets but by high resolution microcopy also reveal compartmentalization (*Wang et al., 2014*). In vivo, aged yeast and mammalian stress granules adopt both liquid and hydrogel-like granule arrangements: they can nucleate as liquid droplets and mature into hydrogels (*Lin et al., 2015*), or are simultaneously comprised of both arrangements with a more solid hydrogel-like core surrounded by a liquid-like shell (*Lin et al., 2015*; *Niewidok et al., 2018*; *Wheeler et al., 2016*).

We are interested in connecting the biophysical properties of *Drosophila* germ granules to their cellular function. Germ granules are part of the germ plasm that forms at the posterior pole during oogenesis where it occupies only ~0.01% of the embryo's volume (*Trcek et al., 2015*). A careful examination of germ plasm with electron microscopy (EM) revealed that germ plasm proteins and mRNAs are organized into small (up to 500 nm) germ granules that are round and non-membrane bound (*Arkov et al., 2006*; *Mahowald, 1962*; *Mahowald et al., 1976*; *Nakamura et al., 1996*). Germ granules are tightly associated with ribosomes indicating that they are sites of active translational regulation. Indeed, referred to as the hubs for post-translational regulation, germ granule localization specifically promotes translation of many germ plasm-enriched mRNAs while their unlocalized counterparts remain translationally repressed (*Gavis and Lehmann, 1994*; *Rangan et al., 2009*). Formation of the germ plasm relies on Oskar protein, whose mRNA localizes at the posterior pole of a developing oocyte. Once translated, the short isoform of Oskar (Short Oskar) recruits other germ plasm components (*Ephrussi and Lehmann, 1992*; *Lehmann, 2016*; *Markussen et al., 1995*). Among these, the core germ plasm protein Vasa, a DEAD-box helicase, Tudor (Tud), the founder of the Tudor domain family of proteins, and Aubergine (Aub), a Piwi family Pi RNA-binding protein (*Lehmann, 2016*), as well as up to 200 maternally-provided mRNAs (*Frise et al., 2010*). A second, N-terminally extended isoform, called Long Oskar, has been implicated in the formation of an extended actin meshwork at the posterior pole (*Tanaka et al., 2011*) where it promotes germ granule tethering (*Rongo et al., 1997*; *Vanzo and Ephrussi, 2002*) and recruits maternally-provided mitochondria (*Hurd et al., 2016*).

Germ plasm is essential for *Drosophila* fertility as it promotes the formation and specification of the PGCs, the first cell lineage to form in the fertilized embryo. At the initial stages of embryonic development, nuclei divide in the center of the embryo. With the onset of the ninth nuclear division nuclei migrate towards the embryo's periphery (*Campos-Ortega and Hartenstein, 1985*; *Su et al., 1998*). Those nuclei that migrate to the posterior end of the embryo become engulfed by the germ plasm. At this stage, germ plasm nuclei become separated from the rest of the embryo by embryonic membranes to form the PGCs, while the remaining nuclei continue their synchronous divisions for four more cycles prior to the cellularization of the soma (*Cinalli and Lehmann, 2013*; *Foe and Alberts, 1983*). Soon after PGCs cellularize, they no longer divide synchronously with the somatic nuclei and arrest in the G2 phase of the cell cycle prior to the onset of germ cell migration (*Cinalli and Lehmann, 2013*; *Su et al., 1998*). In addition to the cytoplasmic germ granules of the germ plasm, nuclear granules appear once the PGCs form (*Jones and Macdonald, 2007*; *Mahowald, 1962*; *Mahowald et al., 1976*). These resemble the cytoplasmic germ granules, but are bigger and appear hollow (*Arkov et al., 2006*; *Mahowald, 1962*; *Mahowald et al., 1976*). These nuclear granules are populated by Oskar (*Jones and Macdonald, 2007*) suggesting that they are germ plasm derived. The function or properties of these granules and their relation to the cytoplasmic granules is unknown.

Here we analyze the biophysical properties of *Drosophila* germ granules in the embryo and find that these granules are best described as phase transitioned condensates. We find that Short Oskar, which nucleates the formation of cytoplasmic germ plasm and germ granules (*Ephrussi and Lehmann, 1992*; *Lehmann, 2016*; *Markussen et al., 1995*), also induces the formation of nuclear

granules in the PGCs. Upon Short but not Long Oskar protein expression in heterologous systems such as *Drosophila* S2R + and mammalian HEK293 cell lines, nuclear granules assemble independently of other germ plasm factors. Vasa is recruited to these granules by Oskar and hence we term these condensates 'nuclear germ granules'. We show that these are phase transitioned granules that display liquid-like and hydrogel-like properties; they can fuse, dissolve, condense and exchange their protein content with the granule environment while they are also stable when purified. Multiple independent domains in Oskar, including its LC and IDR domains synergize to enhance the formation of nuclear germ granules, but surprisingly, no single domain is necessary for granule formation. However, deletion of a nuclear localization sequence (NLS) in Short Oskar ablates nuclear assembly of granules in tissue culture and in the embryo. As a result, PGCs missing nuclear germ granules precociously arrest the cell cycle resulting in a reduced number of PGCs. Together our studies show that *Drosophila* germ granules share properties similar to those of other membraneless granules that form by phase transition. Our studies also reveal a new function for Oskar protein as the nucleator of phase separated granules in the germ cell nuclei. These results provide new insight into how early PGCs division may be regulated independently of the cell cycle timing mechanisms that exist in the syncytial environment of the early embryo.

## Results

### Cytoplasmic germ granules display properties of phase transitioned condensates

The germ granules of the early *Drosophila* embryo morphologically and compositionally resemble other RNP organelles: they are round and membraneless (*Arkov et al., 2006*; *Mahowald, 1962*; *Mahowald et al., 1976*; *Trcek et al., 2015*) and composed of RNA-binding proteins and RNAs (*Hurd et al., 2016*; *Lécuyer et al., 2007*; *Rangan et al., 2009*; *Thomson et al., 2008*; *Voronina et al., 2011*). We analyzed the biophysical properties of germ granules in vivo to probe whether they behave like liquid droplets, where components within the granule are 'liquid-like' and freely exchange with the environment, or phase transitioned granules, which form more stable structures that exchange less readily with their environment. We quantified the biophysical properties of the germ granules during the early nuclear cycles (NC) one to five when germ plasm is organized into cytoplasmic germ granules at the embryo's posterior and the nuclei have not yet reached the poles (*Figure 1Ai–Aiii*). To visualize granules, we used flies that expressed a GFP-tagged Oskar (Osk:GFP) (*Jambor et al., 2015*) and a Kusabira Orange-tagged Vasa (Vasa:KuOr) (*Cinalli and Lehmann, 2013*). Both transgenes spatially and temporally behave like their untagged counterparts (*Trcek et al., 2015*) indicating that they are appropriate germ granule markers. During this stage the two transgenes co-localized within the same granule while they were largely absent from the somatic regions of the embryo (*Figure 1Ai–Aiii*, *Figure 1—figure supplement 1A*), as demonstrated previously (*Trcek et al., 2015*; *Vanzo and Ephrussi, 2002*). Indeed, the concentration of Osk:GFP and Vasa:KuOr was 15 to 21 higher, respectively in granules compared to the surrounding intergranular space or the surrounding somatic cytoplasm (*Figure 1B*, *Figure 1—figure supplement 1B*). Thus, during NC one to five, the bulk of germ plasm activity appears concentrated within granules rather than in the intergranular space.

To assay how granule proteins exchange with the granule environment, we used fluorescent recovery after photobleaching (FRAP). We photobleached Osk:GFP granules and recorded that $43.6 \pm 0.7$ percent of Osk:GFP rapidly exchanged with the granule environment with a half time to complete fluorescent recovery ($t_{1/2}$) of $10.5 \pm 0.9$ s, while the rest of Oskar protein remained associated within the granule (*Figure 1C*, green circles, *Figure 1—video 1*). We recorded a similar FRAP recovery for Vasa:GFP (*Figure 1—figure supplement 1C*). Because the intergranular space was largely devoid of Osk and Vasa (*Figure 1B*), these results suggested that the proteins that re-populated the granules arrived from neighboring granules. To address this question, we used fluorescence loss in photobleaching (FLIP) assay. We continuously bleached a small germ plasm region (region A) (*Figure 1Di*, *Figure 1—video 2*) and recorded the fluctuation of Osk:GFP fluorescence within the bleached region as well as in neighboring, non-bleached regions B, C and D over time (*Figure 1Di*,ii). We detected a significant depletion of fluorescence in region B in immediate proximity to region A (*Figure 1Di*,ii). This depletion of fluorescence was 20 to 30 percent greater than that

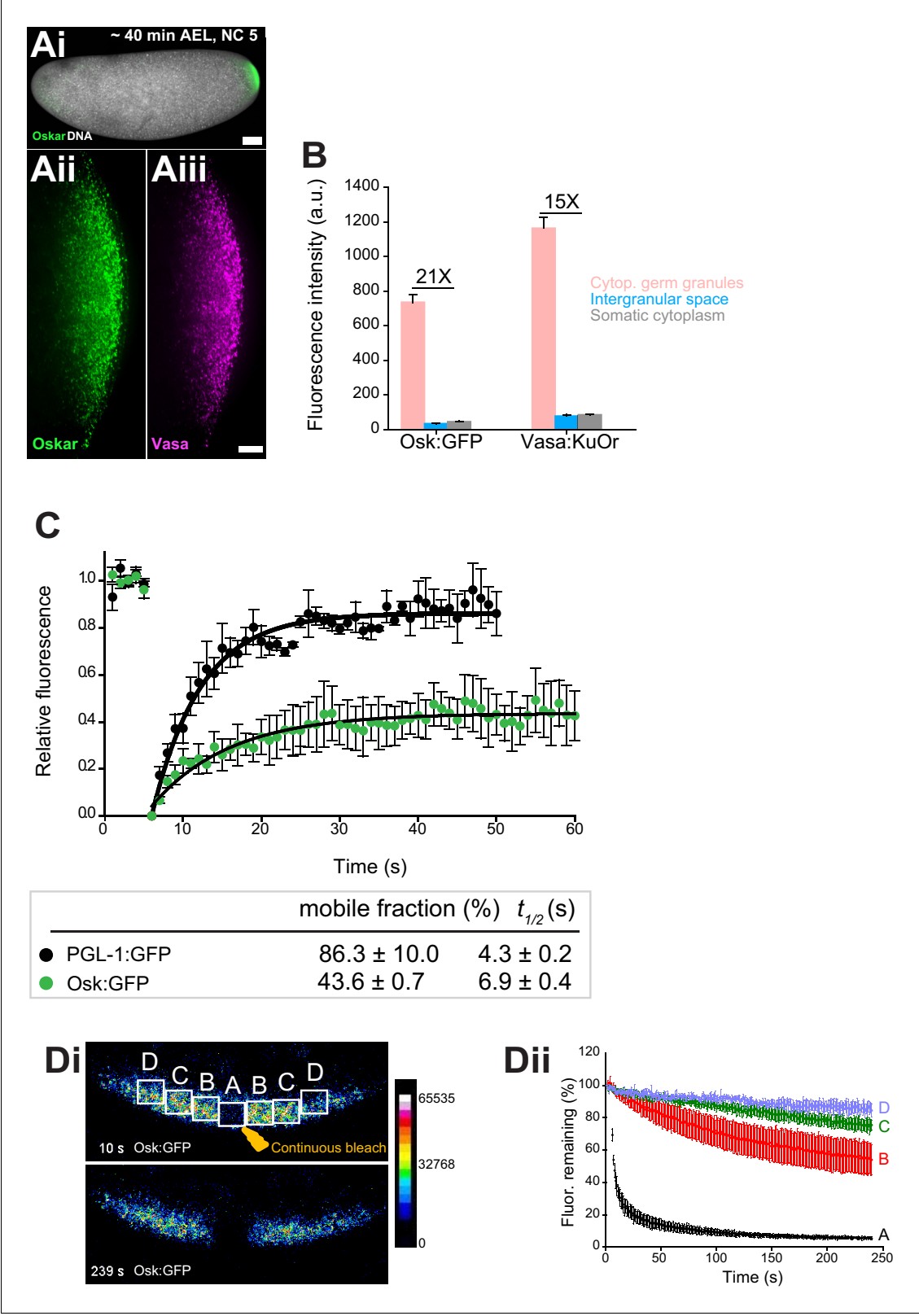

**Figure 1.** Cytoplasmic germ granules display properties of phase transitioned condensates. (**Ai-iii**) *Drosophila* embryos stained with an antibody against Oskar (Ai) (green) and counter-stained for DNA with DAPI (white) or expressing Osk:GFP (green) and Vasa:KuOr (magenta) (ii, iii) at NC five. (**B**) Levels of Osk:GFP and Vasa:KuOr fluorescence in cytoplasmic germ granules (pink bar), the intergranular space (blue bar) and the somatic cytoplasm (grey bar). 21X and 15X fold enrichment of Osk:GFP and Vasa:KuOr fluorescence relative to the intergranular space is marked, respectively. For each

*Figure 1 continued*

bar, mean fluorescent levels per area unit ±SEM of 23 granules, 20 ROIs in the intergranular space and 20 ROIs in somatic cytoplasm are shown (*Figure 1—figure supplement 1B*). (C) FRAP of PGL-1:GFP (one cell zygote in *C. elegans*, black circles) and Osk:GFP located in the cytoplasmic germ granules (green circles). Mean ±SEM of three Osk:GFP ROIs (green) and five P granules (black) is shown. Black lines show fit to the experimental data. Below the graph, the percent mobile fraction and half time to full recovery ($t_{1/2}$) derived from C are shown. (Di,ii). FLIP of Osk:GFP in the early embryo. Region A was continuously bleached for four minutes and an image of the embryo acquired every second. Images in i are pseudo-colored to demonstrate fluorescence intensity levels. ii Fluctation of Osk:GFP fluorescence in regions A,B,C,D (shown in i) and plotted as percent relative to time 0. Normalized mean fluorescence levels ± STDEV of 2 (region A) and 4 (region B-D) ROIs of equal size are shown. Scale bar in Aiii is 10 μm, Ai is 50 μm.

DOI: https://doi.org/10.7554/eLife.37949.002

The following video and figure supplement are available for figure 1:

**Figure supplement 1.** Cytoplasmic germ granules display properties of phase transitioned condensates.
DOI: https://doi.org/10.7554/eLife.37949.003
**Figure 1—video 1.** FRAP of Osk:GFP in cytoplasmic germ granules in embryos.
DOI: https://doi.org/10.7554/eLife.37949.004
**Figure 1—video 2.** Continuous photobleaching of Osk:GFP in germ plasm.
DOI: https://doi.org/10.7554/eLife.37949.005
**Figure 1—video 3.** Continuous photobleaching of an ROI outside of embryo.
DOI: https://doi.org/10.7554/eLife.37949.006
**Figure 1—video 4.** Continuous photobleaching of Vasa:GFP in germ plasm.
DOI: https://doi.org/10.7554/eLife.37949.007
**Figure 1—video 5.** Continuous photobleaching of an ROI outside of embryo.
DOI: https://doi.org/10.7554/eLife.37949.008
**Figure 1—video 6.** FRAP of PGL-1:GFP in *C. elegans* P granules.
DOI: https://doi.org/10.7554/eLife.37949.009

observed in the distant regions C and D, respectively (*Figure 1Di, ii*), as well as 30 percent greater than would be anticipated if the depletion of fluorescence was due to unintentional photobleaching during imaging (*Figure 1—figure supplement 1Di,ii, Figure 1—video 3*). We detected a similar behavior for Vasa:GFP (*Figure 1—figure supplement 1Ei,ii, Fi,ii, Figure 1—video 4* and *5*). Importantly, the fraction of mobile Osk:GFP and Vasa:GFP recorded by FRAP (43.6% and 46.0%, respectively) was similar to that recorded by FLIP (~30%) suggesting that the same mobile fraction was captured by both assays. The redistribution of fluorescence was not due to the movement of granules among regions as these movements appeared corralled rather than directional (*Figure 1—video 1* and *3*), are infrequent and occur over a much longer time period (*Sinsimer et al., 2013*). Rather, Osk:GFP and Vasa:GFP dynamically exchanged among neighboring granules. However, despite displaying liquid-like properties, we and others have not observed fusion of cytoplasmic germ granules (*Sinsimer et al., 2013*), as had been described for P granules in *C. elegans* (*Brangwynne et al., 2009*). To probe this question further, we compared the mobility of Osk:GFP and Vasa:GFP with the mobility of PGL-1, a core constituent of P granules found in a one-cell zygote of *C. elegans* (*Kawasaki et al., 1998*). We photobleached PGL-1:GFP and recorded diffusion kinetics similar to published reports (86.3 ± 10.0 percent mobile fraction with $t_{1/2}$ of 4.3 ± 0.2 s; *Figure 1C*, black circles, *Figure 1—video 6*, *Figure 4—figure supplement 1C*) (*Brangwynne et al., 2009*). Thus, unlike highly mobile P granules in *C. elegans* that form during LLPS (*Brangwynne et al., 2009*), cytoplasmic germ granules in *Drosophila* appear comprised of liquid and more hydrogel-like states and are thus best described as phase transitioned condensates.

## Core germ granule proteins Oskar and Vasa form phase transitioned nuclear germ granules in primordial germ cells

In the wild-type embryos, PGC form at nuclear cycle 10, after the syncytial nuclei reach the cortex and form actin filled membrane protrusions. In the germ plasm region, each such protrusion, called pole bud, generates two PGCs through the coordinated action of two orthogonally placed constrictions: one constriction is the consequence of mitotic anaphase, the other is the result of an unusual, microtubule-independent cleavage (*Campos-Ortega and Hartenstein, 1985*; *Cinalli and Lehmann, 2013*; *Pae et al., 2017*). During every division of the newly formed PGCs the cytoplasmic germ granules were associated with the pericentriolar region (*Figure 2Ai–Aiii*) (*Lerit and Gavis, 2011*) but

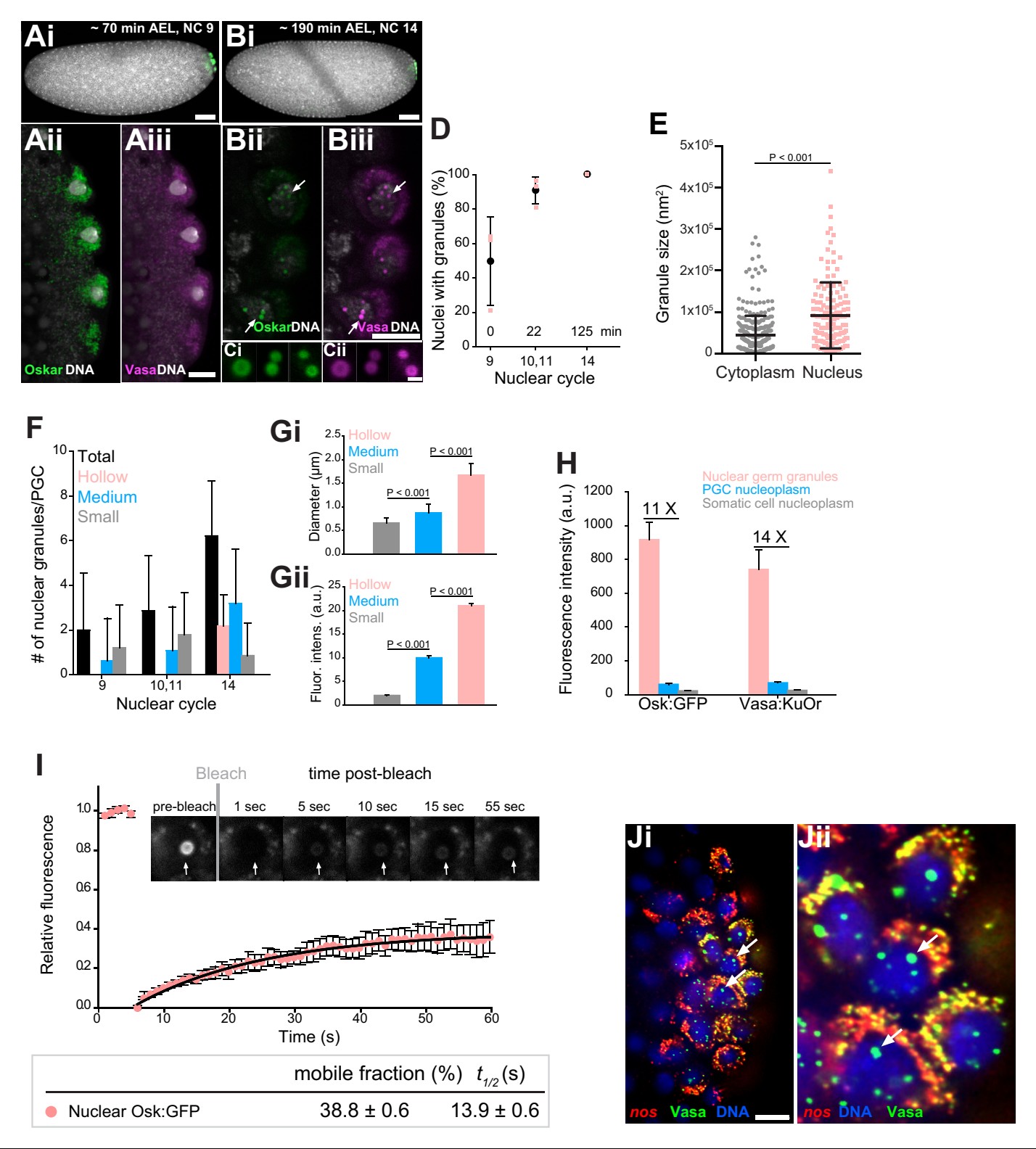

**Figure 2.** Core germ granule proteins Oskar and Vasa form phase transitioned nuclear germ granules in primordial germ cells. (A-B) *Drosophila* embryos stained with an antibody against Oskar (Ai,Bi) (green) and counter-stained for DNA with DAPI (white) or expressing Osk:GFP (green) and Vasa: KuOr (magenta) (Aii,Aiii, Bii,Biii) and counter stained with DAPI (white) at NC nine and 14. White arrows point at hollow nuclear germ granules. (C) Close-up of nuclear germ granules marked by Osk:GFP (i, green) and Vasa:KuOr (ii, magenta) at NC 14. (D) Appearance of nuclear granules in PGCs at NC nine (0 min), NC 10,11 (22 min) and NC 14 (125 min). The number of PGCs with nuclear germ granules was counted at each NC. Three embryos per

*Figure 2 continued on next page*

Figure 2 continued

NC were analyzed and an average percent of granule-containing nuclei per embryo per NC calculated (pink circles). Mean ±STDEV is shown. (E) Nuclear germ granules are larger than cytoplasmic germ granules (91797 nm$^2$ = 606.0 nm diameter vs. 44533 nm$^2$ = 422.1 nm diameter, respectively; unpaired t-test, p<0.0001). Mean ±STDEV is shown. (F) Number of nuclear granules per PGC through early embryogenesis. PGCs of two Vasa:GFP expressing embryos per NC were analyzed and mean ±STDEV of total (black bars), small (grey), medium (blue) and hollow (pink) number of nuclear germ granules per PGC per NC determined. Gi,ii Small, medium and hollow nuclear granules differ in their size (i) and amount of protein (ii) (statistical significance: two-tailed t-test). Mean ±STDEV of 20 to 31 granules is shown. (H) Levels of Osk:GFP and Vasa:KuOr fluorescence in nuclear germ granules (pink bar), in the PGC nucleoplasm (blue bar) and in the somatic cell nucleoplasm (grey bar). 11X and 14X fold enrichment of Osk:GFP and Vasa:KuOr fluorescence relative to the PGC nucleoplasm, respectively, is shown. For each bar, mean fluorescent levels ± SEM of nine granules, 12 ROIs in the PGC nucleoplasm and 15 ROIs in somatic cell nucleoplasm are shown (see *Figure 1—figure supplement 1B*). (I) FRAP of nuclear Osk:GFP germ granules in PGCs. Mean ±SEM of eight hollow nuclear germ granules is shown. Black line shows the fit to the experimental data. Below the graph, the % mobile fraction and $t_{1/2}$ derived from I is shown. Images in the graph show fluorescence recovery before and after photobleaching. White arrow points at the bleached granule. (J) i,ii smFISH reveals that germ plasm mRNA *nos* (red) is enriched in cytoplasmic, but not in nuclear germ granules (green). DNA stained with DAPI is shown in blue. ii close-up of Ji. Arrows point at nuclear germ granules lacking *nos* smFISH signal. Scale bar in Cii is 1 μm and in Aiii, Biii, Ji is 10 μm and in Ai, Bi is 50 μm.

DOI: https://doi.org/10.7554/eLife.37949.010

The following video and figure supplement are available for figure 2:

**Figure supplement 1.** Core germ granule proteins Oskar and Vasa form phase transitioned nuclear germ granules in primordial germ cells.

DOI: https://doi.org/10.7554/eLife.37949.011

**Figure 2—video 1.** FRAP of Osk:GFP in nuclear germ granules of PGCs.

DOI: https://doi.org/10.7554/eLife.37949.012

retained similar size and morphology to the cytoplasmic germ granules present earlier (*Figure 1Ai–Aiii*). As soon as PGCs formed, we started to observe nuclear granules, which grew in size and number, and became hallmarks of PGCs at the cellular blastoderm stage (NC14), when not only the PGCs but also the somatic cells have formed (*Figure 2Ai–Aiii*;Bi-Biii, 2D). Vasa:KuOr co-localized with Osk:GFP in 95.1% of nuclear germ granules (*Figure 2—figure supplement 1A*) and, as we have shown previously, in 90.3% of cytoplasmic germ granules (*Trcek et al., 2015*). Importantly, we did not observe Vasa:KuOr nuclear granules without Osk:GFP (*Figure 2—figure supplement 1A*), indicating that Vasa can populate nuclear germ granules only in the presence of Oskar protein. This observation was consistent with how these proteins behave in cytoplasmic germ granules; only Oskar nucleates cytoplasmic germ granules (*Ephrussi and Lehmann, 1992*; *Markussen et al., 1995*) and recruits Vasa to granules where the two also physically interact (*Breitwieser et al., 1996*; *Lehmann, 2016*).

On average, the nuclear granules were two-times bigger than the cytoplasmic granules (*Figure 2E*). Initially, mostly small granules with a diameter of 0.7 ± 0.1 μm were present and were similar in size to cytoplasmic germ granules (*Figure 2E,F,Gi*; *Figure 1—figure supplement 1Bi*). Over time, they became more numerous, grew in size and accumulated more protein (*Figure 2F–Gii*; *Figure 2—figure supplement 1B–C*). Toward the end of the NC 14, characteristic hollow granules with a diameter of 1.7 ± 0.3 μm appeared (*Figure 2F,Gi*). As with cytoplasmic germ granules, we could not detect fusion of nuclear germ granules. Nuclear granules did not result from over-expression of Osk:GFP or Vasa:KuOr since they have been previously observed in wild-type flies using EM and immunofluorescence (*Jones and Macdonald, 2007*; *Mahowald, 1962*; *Mahowald et al., 1976*). Because nuclear granules formed only in PGCs (*Figure 2—figure supplement 1B*) and were populated by Oskar and Vasa (*Figure 2Ci–ii*), we termed these condensates 'nuclear germ granules'.

Nuclear germ granules highly concentrated Osk:GFP and Vasa:KuOr while the surrounding nucleoplasm contained 11 to 14 fold less protein, respectively, similar to the levels found in the somatic nucleoplasm (*Figure 2H*, *Figure 2—figure supplement 1D*). Furthermore, FRAP kinetics of the condensed Osk:GFP and Vasa:GFP were similar to those recorded for these two proteins in cytoplasmic germ granules (38.8 ± 0.6% mobile fraction, $t_{1/2}$ of 13.9 ± 0.6 s and 51.1 ± 0.5%, $t_{1/2}$ of 8.7 ± 0.7 s, respectively (*Figure 2I*, *Figure 1C*, *Figure 2—video 1*, *Figure 2—figure supplement 1E*)), indicating that the biophysical properties of cytoplasmic and nuclear Oskar and Vasa were similar.

Finally, we asked if nuclear germ granules accumulated mRNAs similar to those found in cytoplasmic germ granules. Using in vitro RNA-binding assays, Oskar was shown to specifically bind *nanos* (*nos*), *polar-granule-component* (*pgc*) and *germ-cell-less* (*gcl*) mRNAs (three highly germ

granule-enriched and maternally-deposited mRNAs [*Little et al., 2015*; *Trcek et al., 2015*]), while it can also interact with a variety of RNAs with lower affinity (*Jeske et al., 2015*; *Yang et al., 2015*). This finding suggested that Oskar could directly enrich mRNAs in cytoplasmic germ granules, the hubs for post-transcriptional mRNA regulation. Using single-molecule fluorescence in situ hybridization (smFISH) (*Trcek et al., 2015*; *Trcek et al., 2017*) we found that *nos*, *pgc* and *gcl* did not localize to the nucleus or nuclear germ granules (*Figure 2Ji,Jii*, white arrows; *Figure 2—figure supplement 1Fi-vi*). This and the fact that other cytoplasmic granule components such as Aub and Tud do not associate with nuclear granules (see below, *Figure 3—figure supplement 1H*) suggest that nuclear germ granules may function differently from cytoplasmic germ granules.

In summary, nuclear germ granules are round and membraneless structures, populated by Oskar and Vasa (*Jones and Macdonald, 2007*; *Mahowald, 1962*; *Mahowald et al., 1976*) and behave like phase transitioned condensates. Importantly, morphologically, biophysically and at least in part by composition they resemble cytoplasmic germ granules indicating that the mechanism that drives their formation might be shared.

## Expression of short Oskar in cell lines reconstitutes nuclear germ granules

Depletion of germ granule components such as Oskar and Vasa prevents the assembly of germ granules during oogenesis. As a result, embryos laid by *osk* or *vas* mutant mothers fail to form PGCs (*Lehmann and Nüsslein-Volhard, 1986*; *Thomson and Lasko, 2004*; *Thomson et al., 2008*), precluding the observation of nuclear germ granules. To study the properties and functions of nuclear germ granules free of these complications and given the role of Oskar as the nucleator of germ plasm (*Ephrussi and Lehmann, 1992*; *Lehmann, 2016*; *Markussen et al., 1995*), we asked whether Oskar could assemble granules in *Drosophila* S2R+ cells. These widely used cultured cells were derived from embryonic soma and do not express core germ plasm proteins (*modENCODE Consortium et al., 2010*). Translation of *osk* mRNA produces two isoforms, long and short (Long Osk, Short Osk, respectively) that differ in the first 138 N-terminal aa (*Markussen et al., 1995*). Only Short, but not Long Osk is necessary and sufficient to instruct the formation of cytoplasmic germ granules (*Breitwieser et al., 1996*; *Markussen et al., 1995*; *Vanzo and Ephrussi, 2002*). Consistently, when transiently expressed in S2R+ cells, Short Osk tagged at its N-terminus with mCherry (Short mCherry:Osk) organized round, membraneless and often hollow nuclear germ granules, while Long Osk did not (*Figure 3A–C Figure 3—figure supplement 1Ai,Aii*) (also reported by [*Jeske et al., 2017*]). This approach allowed for a highly controlled introduction of germline proteins into cell lines *via* transient transfection allowing identification of minimal components required for nuclear germ granule formation.

Three types of Short mCherry:Osk granules were observed that differed in size, protein abundance and morphology: small, hollow, and big and non-hollow (*Figure 3D–F*). The former two resembled nuclear germ granules described in PGCs (*Figure 2B,Ci*) (*Jones and Macdonald, 2007*; *Mahowald, 1962*; *Mahowald et al., 1976*), while the latter were observed only in S2R+ cells. Importantly, the formation of these granules depended on Short Osk and not on the mCherry fluorophore or the orientation of the fluorescent tag, as N- and C-terminally-tagged Short Osk formed the same granules (*Figure 3—figure supplement 1B–D*). Furthermore, the ability of Short Osk to organize into nuclear germ granules was not limited to *Drosophila* cells; when driven by the EF-1α promoter, Short Osk:mCherry formed nuclear granules in HEK293 cells, a human embryonic kidney cell line devoid of *Drosophila* proteins (*Figure 3G*). We conclude that Short Oskar is able to assemble into nuclear granules in heterologous cell systems independent of other germ plasm proteins.

To test whether the ability to form nuclear granules is unique to Short Osk, we transfected S2R + cells with other germ plasm proteins. We tested a subset of the 119 proteins that we previously identified in a Short Osk immunoprecipitation followed by mass spectroscopy (*Hurd et al., 2016*). These included core granule components Vasa, Tud and Aub (*Arkov et al., 2006*; *Hurd et al., 2016*; *Voronina et al., 2011*) and other known granule interactors, Piwi, DCP1 and Cup (*Hurd et al., 2016*; *Voronina et al., 2011*), as well as 113 previously unknown germ granule constituents (*Figure 3—source data 1*) (*Arkov et al., 2006*; *Gao and Arkov, 2013*; *Hurd et al., 2016*; *Thomson et al., 2008*; *Voronina et al., 2011*). A gene ontology (GO) term analysis revealed that many proteins identified were involved in germ cell development and post-transcriptional gene regulation including several newly identified mRNA binding proteins, RNP binding proteins and ATP

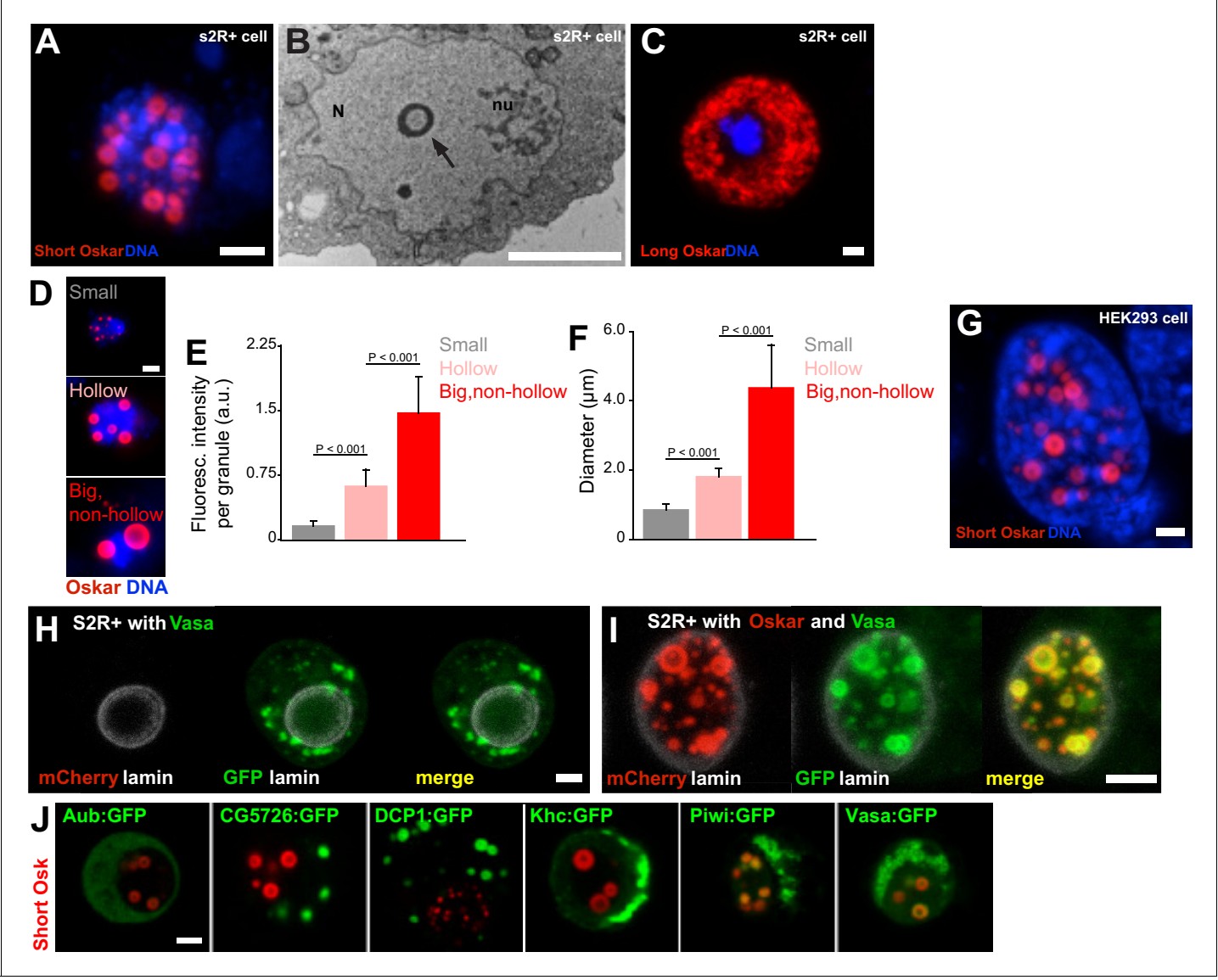

**Figure 3.** Expression of Short Oskar in cell lines reconstitutes nuclear germ granules . (**A**) Short mCherry:Osk forms hollow nuclear granules when transfected into cultured *Drosophila* S2R+ cells. (**B**) Transmission electron micrograph of a *Drosophila* cell transfected with Oskar shows the cross-section of a granule (arrow) with an electron-lucid core located within the nucleus (N) and separate from the nucleolus (nu). (**C**) Long Oskar does not form granules in S2R+ cells. (**D-F**) Three types of Short mCherry:Osk nuclear germ granules can be observed in S2R+ cells (small, hollow and big and non-hollow) that differ in the amount of protein (**E**) and in their size (**F**). In E and F, mean ± STDEV of 10 (small, hollow) and five (big, non-hollow) granules is shown (statistical significance: two-tailed t-test). (**G**) Short Osk:mCherry forms nuclear granules when transfected into human HEK293 cells. (**H**) S2R+ cell transiently transfected with a plasmid expressing Vasa:GFP (green) or, (**I**), co-transfected with Short mCherry:Osk (red) and Vasa:GFP (green) and counter-stained with an antibody for nuclear lamin (white). (**J**) S2R+ cells co-transfected with Short mCherry:Osk (red) and Aub:GFP, CG5726:GFP, DCP1:GFP, Khc:GFP, Piwi:GFP or Vasa:GFP (green). Scale bar in all is 2 μm.

DOI: https://doi.org/10.7554/eLife.37949.013

The following source data and figure supplement are available for figure 3:

**Source data 1.** We previously identified proteins associated with Short Osk from early embryos using IP/mass spec (*Hurd et al., 2016*).
DOI: https://doi.org/10.7554/eLife.37949.015

**Figure supplement 1.** Expression of Short Oskar in cell lines reconstitutes nuclear germ granules.
DOI: https://doi.org/10.7554/eLife.37949.014

binding proteins such as RNA helicases (*Figure 3—figure supplement 1E*, *Figure 3—source data 1*). Since RNA binding proteins and RNA helicases are known components of non-membrane-bound protein granules and can even prompt their formation (*Courchaine et al., 2016*; *Decker and Parker, 2012*; *Voronina et al., 2011*), we reasoned that some of these proteins could self-organize into granules independently of Short Osk when expressed in S2R+ cells. To test this hypothesis, we focused on known germ granule components: Vasa, a DEAD-box RNA helicase (*Hay et al., 1988*) and two Piwi family RNA-binding proteins Piwi and Aub (*Harris and Macdonald, 2001*; *Juliano et al., 2011*), as well as several proteins highly abundant in short-Osk IP: DCP1, an mRNA decapping protein involved in mRNA turnover (*Beelman et al., 1996*; *Lin et al., 2006*), Kinesin heavy chain protein Khc, a microtubule motor protein that functions in the long-distance transport of cytoplasmic cargoes such as *osk* mRNA (*Brendza et al., 2000*) and a protein of unknown biological function encoded by the CG5726 gene.

When expressed in S2R+ cells, none of these proteins was capable of forming nuclear granules on their own (*Figure 3H*, *Figure 3—figure supplement 1G*). Vasa:GFP condensed into cytoplasmic granules when expressed alone (*Figure 3H*), consistent with the observations that Ddx4, a mouse homolog of Vasa, forms round and non-membrane-bound cytoplasmic aggregates both in vivo and in vitro (*Nott et al., 2015*). However, co-expression of Vasa:GFP with mCherry:Osk readily stimulated its incorporation into nuclear granules (*Figure 3I,J*) (*Jeske et al., 2017*), where 80.3% of granules were populated by both proteins. 7.2% of Vasa positive granules appeared not to contain Oskar (*Figure 3—figure supplement 1F*). However, since Vasa cannot form nuclear germ granules in the absence of Oskar, we assume that the small number of Vasa nuclear germ granules lacking Oskar are populated by both proteins but at various amounts rather than Vasa forming distinct granules devoid of Oskar protein.

Co-expression of Piwi:GFP with Short mCherry:Osk also stimulated its incorporation into nuclear granules (*Figure 3J*, *Figure 3—figure supplement 1G*), while in the embryo Piwi accumulated in the nuclei of both somatic and germ cells (*Figure 3—figure supplement 1H*). In contrast, Aub despite its co-localization with Osk and Vasa in cytoplasmic germ granules (*Trcek et al., 2015*) never relocated with Short Osk into the nuclei of either S2R+ cells (*Figure 3J*, *Figure 3—figure supplement 1G*) consistent with its behavior in PGCs (*Figure 3—figure supplement 1H*), as previously observed (*Jones and Macdonald, 2007*) (*Figure 3—figure supplement 1H*). Rather, in S2R+ cells Aub remained diffuse in the cytoplasm (*Figure 3—figure supplement 1G*) and like Tud, that recruits Aub to germ granules (*Liu et al., 2010*) is granule-bound in the cytoplasm of PGCs (*Figure 3—figure supplement 1H*).

Finally, while unable to form nuclear granules alone or when co-expressed with Short Osk, CG5726 and DCP1 formed distinct round-shaped cytoplasmic granules in the absence of Short Osk (*Figure 3J*, *Figure 3—figure supplement 1I*). While the function of CG5726 is unknown, the biology of DCP1 is very well understood and this behavior of DCP1 is consistent with observations in other organisms and systems; DCP1 is a core component of the yeast and mammalian P bodies where it regulates mRNA stability through mRNA decapping (*Beelman et al., 1996*; *Decker and Parker, 2012*), it also affects localization of *oskar* mRNA in the oocyte and populates *Drosophila* sponge bodies (*Lin et al., 2006*), mouse chromatoid bodies (*Kotaja et al., 2006*) the perinuclear nuage of mouse spermatocytes (*Aravin et al., 2009*) and the P granule-associated P bodies in *C. elegan*s (*Gallo et al., 2008*). Interestingly, in S2R+ cells, GFP-tagged DCP1 formed spherical, cytoplasmic granules with a protein-lucid core (*Figure 3—figure supplement 1I*) similar to the ones formed by Osk in the embryo and in cell lines (*Figure 2Ci*, *Figure 3A,G*) suggesting that the biophysical properties driving the formation of Short Osk and DCP1 granules may be shared between the two proteins.

Our results reveal that Short Osk forms nuclear granules in S2R + and HEK293 cells. Cell line-expressed Short Osk recapitulated properties of nuclear granules observed in the embryos such as granule morphology and the ability to organize other germ granule components. Together, these results suggest that this system may provide an excellent model to study the molecular and biophysical properties of nuclear germ granules. This is particularly important as Short Osk is insoluble during purification (*Yang et al., 2015*), which would preclude studying the ability of this protein to phase separate in a test tube (*Li et al., 2012*).

## Nuclear germ granules in S2R+ cells are phase transitioned condensates

To determine whether the cell culture system indeed recapitulated other essential features of nuclear germ granules we characterized their biophysical properties in S2R+ cells. Similar to the measurements obtained on nuclear germ granules formed in PGCs (*Figure 2H*), we found that the majority of the Short mCherry:Osk fluorescence was located in granules rather than in the nucleoplasm (*Figure 4A*, *Figure 4—figure supplement 1A*). As the levels of nuclear Short mCherry:Osk increased (*Figure 4B*), the total number and the number of small nuclear germ granules decreased and the number of hollow and big and non-hollow granules increased (*Figure 4C*), indicating that Short mCherry:Osk granules may fuse. Indeed, rapid live cell imaging of transfected S2R+ cells revealed that nuclear germ granules did fuse (*Figure 4D*, *Figure 4—video 1*) albeit slower and less frequently than P granules in *C. elegans* (*Figure 4—video 2*) (*Brangwynne et al., 2009*).

FRAP experiments revealed that in S2R+ cells 46.8% of nuclear Short mCherry:Osk readily exchanged with the granule environment (*Figure 4E,F*, *Figure 4—video 3*). The kinetics of recovery after photobleaching were similar to those recorded for Osk:GFP in nuclear and cytoplasmic germ granules in the embryos (*Figure 4E,F*) and comparable to those recorded for P granule proteins (*Brangwynne et al., 2009*; *Sheth et al., 2010*), stress granule proteins (*Kedersha et al., 2000*), P body proteins (*Aizer et al., 2008*; *Andrei et al., 2005*) and Balbiani body proteins (*Boke et al., 2016*) (*Figure 4—figure supplement 1C*). When granules were only partially bleached, 75.4% of mCherry:Osk fluorescence exchanged during FRAP (*Figure 4—figure supplement 1B*), suggesting that the recovery of Oskar protein resulted from internal rearrangement of mCherry:Osk as well as its exchange with the granule environment. Indeed, photoconversion of Dendra2-tagged Short Osk (*Figure 4G*, white arrows, *Figure 4—video 4*) revealed that a fraction of Short Dendra2:Osk readily exchanged among neighboring granules (*Figure 4G*, red arrows, *Figure 4—video 4*). Together these results indicate that in S2R+ cells a fraction of nuclear Short Osk actively condensed into granules and dissolved, a behavior similar to cytoplasmic germ granules in the embryo (*Figure 1C–D*) and to P granules in *C. elegans* (*Brangwynne et al., 2009*; *Sheth et al., 2010*). Despite the ability to exchange, less than 50 percent of Oskar was mobile in the three types of granules analyzed (cytoplasmic and nuclear germ granules in the embryo and nuclear germ granules in S2R+ cells) (*Figure 4E,F*), suggesting that *Drosophila* cytoplasmic and nuclear germ granules appear more stable than other phase separated granules (*Figure 4—figure supplement 1C*). Indeed, Short Dendra2:Osk granules isolated from S2R+ cells could still be observed one hour after purification and once photoconverted they maintained their fluorescence levels (*Figure 4H,I*). Furthermore, aliphatic alcohols 1,6 hexanediol and 1,5-pentanediol that disrupt weak hydrophobic interactions (*Kroschwald et al., 2015*; *Lin et al., 2016*; *Patel et al., 2007*) and cause rapid disassembly of liquid droplets but not of hydrogels (*Kroschwald et al., 2015*; *Lin et al., 2016*; *Wheeler et al., 2016*) did not dissolve nuclear germ granules formed in S2R+ cells even at high concentrations (*Figure 4—figure supplement 1D*). Thus, a fraction of the nuclear germ granule formed in S2R+ cells likely assumed a more stable, hydrogel-like conformation.

Our experiments show that S2R+ cells recapitulated the essential features of the nuclear germ granules formed in PGCs (their morphology, composition, biophysical properties and cellular localization) making this a useful system to study the assembly and function of nuclear germ granules outside of *Drosophila*. Our biophysical studies revealed that like in the embryo, nuclear germ granules in S2R+ cells are phase transitioned condensates. While a portion of the granule protein adopted liquid-like behavior, a fraction also exhibited hydrogel-like immobility, suggesting a more structured organization of these granules. To gain better understanding of this process, we next sought to determine the functional domain(s) of Short Oskar that can promote its condensation and localization to the nucleus.

## Multiple independent Oskar domains synergize to promote granule formation

A proposed signature of proteins that condense are LC sequences or IDRs, as these domains are thought to facilitate non-specific protein-protein interactions (*Banani et al., 2016*; *Courchaine et al., 2016*; *Lin et al., 2015*; *Protter et al., 2018*). In test tube assays, proteins that phase separate often contain RNA binding domains and require these for optimal aggregation, and may also harbor other more specific protein-protein interaction domains that can promote

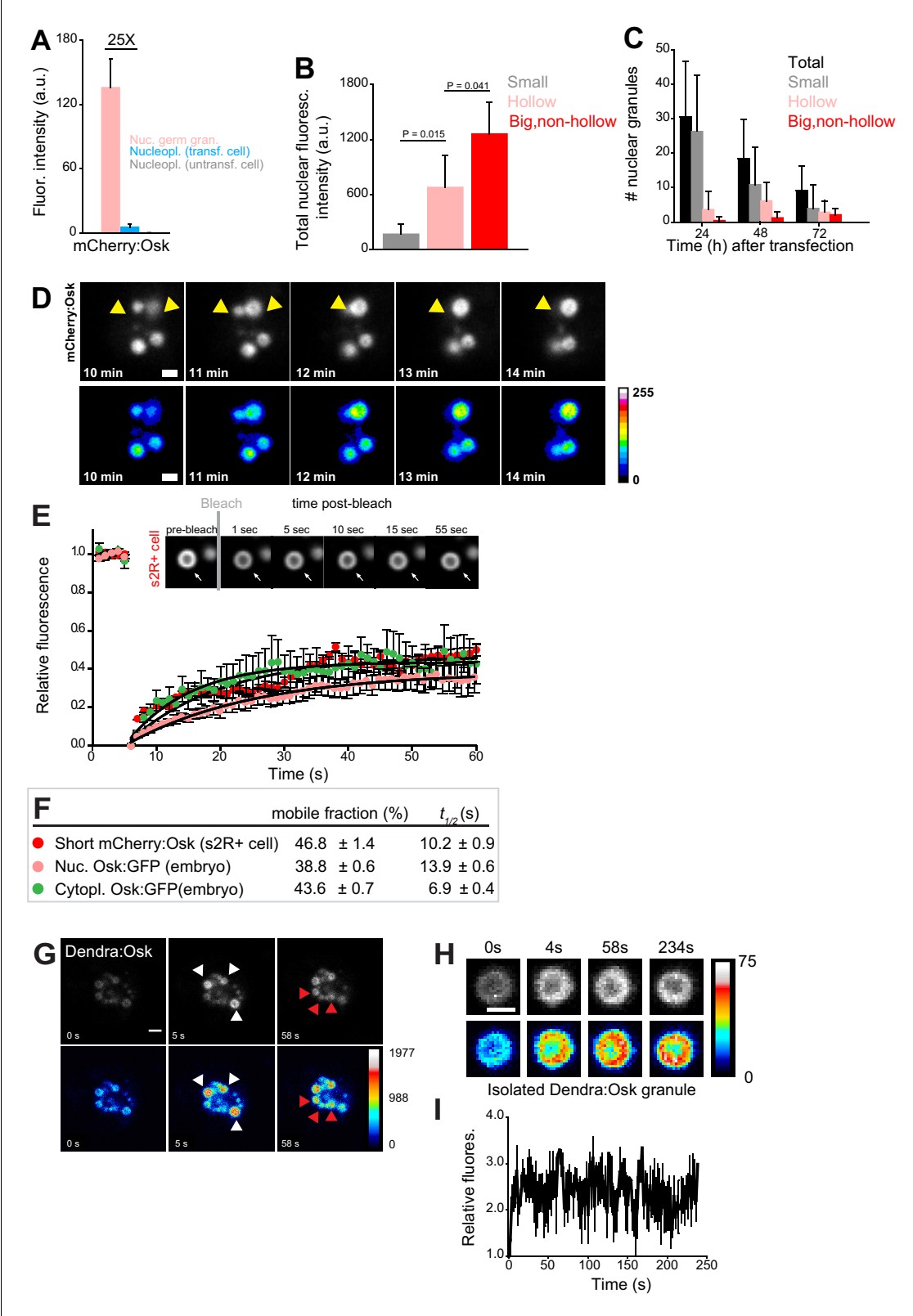

**Figure 4.** Nuclear germ granules in S2R+ cells are phase transitioned condensates. (A) Levels of Short mCherry:Osk fluorescence in nuclear germ granules (pink bar), in the nucleoplasm of transfected (blue bar) and of untransfected (grey bar) cells. 25 fold enrichment of granular vs. nucleoplasmic Short mCherry:Osk fluorescence is shown. For each bar, mean fluorescence levels ± STDEV are shown where 18 granules and 20 ROIs located in the nucleoplasm of transfected and untransfected cells each are shown (statistical significance: two-tailed t-test). (B) Total fluorescence intensity of nuclear

*Figure 4 continued on next page*

*Figure 4 continued*

Short mCherry:Osk in cells that accumulate small, hollow and big, non-hollow nuclear germ granules (statistical significance: two-tailed t-test). Mean ± STDEV of five (small, hollow) and four (big, non-hollow) nuclei per granule type is shown. (C) Number of nuclear germ granules in S2R+ cells over time. Mean ±STDEV number of the total (black), small (grey), hollow (pink) and big,non-hollow (red) granules per nucleus per time point is shown. Per time point, 20 cells were analyzed. (D) Fusion of Short mCherry:Osk nuclear granules (arrow heads) in S2R+ cells. Pseudo-colored images demonstrate fluctuations of fluorescence intensity. (E,F) FRAP of nuclear Short mCherry:Osk in S2R+ cells (red circles). Mean ±SEM of eight hollow nuclear germ granules is shown. Black line shows the fit to the experimental data. Images in the graph show FRAP recovery of a nuclear granule in S2R + cell before and after photobleaching. Also shown are FRAP of Osk:GFP located in cytoplasmic germ granules (green circles, *Figure 1C*) and in nuclear germ granules (pink circles, *Figure 2I*) with accompanying % mobile fraction and $t_{1/2}$. (G) Photoconversion of three Short Dendra2:Osk granules (white arrowheads) at t = 5 s demonstrates that Short Osk dynamically exchanges among neighboring nuclear germ granules (red arrowheads) in S2R + cells. (H) Photoconversion of isolated Short Dendra2:Osk granules one hour after their purification. Raw and pseudo-colored images of a hollow granule are shown before (0 s) and after (4 s-234s) photoconversion. I Fluorescence intensity (mean ± SEM) of three isolated Short Dendra2:Osk granules before (0 s) and after (4 s-234s) photoconversion. Scale bar in all is 2 μm.

DOI: https://doi.org/10.7554/eLife.37949.016

The following video and figure supplement are available for figure 4:

**Figure supplement 1.** Nuclear germ granules in S2R+ cells are phase transitioned condensates.

DOI: https://doi.org/10.7554/eLife.37949.017

**Figure 4—video 1.** Fusion of Short mCherry:Osk nuclear germ granules in S2R+ cells.

DOI: https://doi.org/10.7554/eLife.37949.018

**Figure 4—video 2.** Not all Short mCherry:Osk nuclear germ granules fuse in S2R+ cells.

DOI: https://doi.org/10.7554/eLife.37949.019

**Figure 4—video 3.** FRAP of Short mCherry:Osk in nuclear germ granules in S2R+ cells.

DOI: https://doi.org/10.7554/eLife.37949.020

**Figure 4—video 4.** Short Dendra2:Osk nuclear germ granules continuously condense and dissolve in S2R+ cells.

DOI: https://doi.org/10.7554/eLife.37949.021

oligomerization (*Banani et al., 2016*). Short Osk is composed of two known structured domains located at the N- and C-terminal ends of the protein, called LOTUS and SGNH, respectively (*Figure 5A*) (*Jeske et al., 2015*; *Yang et al., 2015*). The LOTUS domain is responsible for Oskar's ability to homodimerize and to specifically associate with Vasa (*Jeske et al., 2017*). The first 47 aa of the LOTUS domain also harbor a predicted LC sequence and were not characterized in the recent crystal structure of the Oskar dimer in association with Vasa (*Figure 5B*, orange box) (*Jeske et al., 2017*). The SGNH region, which is structurally related to hydrolases, forms a novel RNA-binding domain (*Jeske et al., 2015*; *Yang et al., 2015*). LOTUS and SGNH are connected by a 160 aa linker that is predicted to form an IDR (*Figure 5B*, green box) but was also shown to bind LASP, an actin binding protein and Valois, a component of the PRMT5 methyltransferase complex (*Ahuja and Extavour, 2014*; *Anne and Mechler, 2005*). Based on the predicted binding sites for these two proteins, we divided the linker region into region L1 and L2, respectively (*Figure 6A*).

To determine, whether any single region of Short Osk was required for condensation into granules, we individually deleted each of the four domains and transiently expressed them as mCherry-tagged truncation constructs in S2R+ cells. We found that none of the four regions was required for granule formation per se (*Figure 5C–G*). Consistent with the known biological function of the LOTUS domain in Vasa binding, Osk-ΔLOTUS failed to localize Vasa to nuclei and instead Vasa took on a diffuse and non-granular distribution in S2R+ cells that resembled its distribution in the absence of Oskar (*Figure 5—figure supplement 1B*) (*Jeske et al., 2017*). Osk-ΔL1 protein did not localize to the nucleus, but instead accumulated in cytoplasmic granules and Osk-ΔL2 granules were found in the cytoplasm and the nucleus. Surprisingly, deletion of L2 prevented accumulation and co-localization of Vasa:GFP with Osk-ΔL2 protein specifically in the nucleus, while it remained co-localized, albeit faintly, with Osk-ΔL2 in cytoplasmic foci (*Figure 5—figure supplement 1B*). The size of nuclear granules formed by Osk-ΔLOTUS, Osk-ΔL1 and Osk-ΔL2 mutants was similar to those formed by full length Short Osk, while deletion of the SGNH domain lead to the formation of much larger nuclear granules, which lacked the appearance of a hollow core (*Figure 5H*, *Figure 5—figure supplement 1A*).

Together our results suggest that the ability of Short Osk to homo-dimerize, interact with Vasa, Valois and LASP or bind RNA is dispensable for nuclear granule formation, nor were the LC sequence or the IDR necessary (*Figure 5C–H*, *Figure 5—figure supplement 1A*). Thus, the four

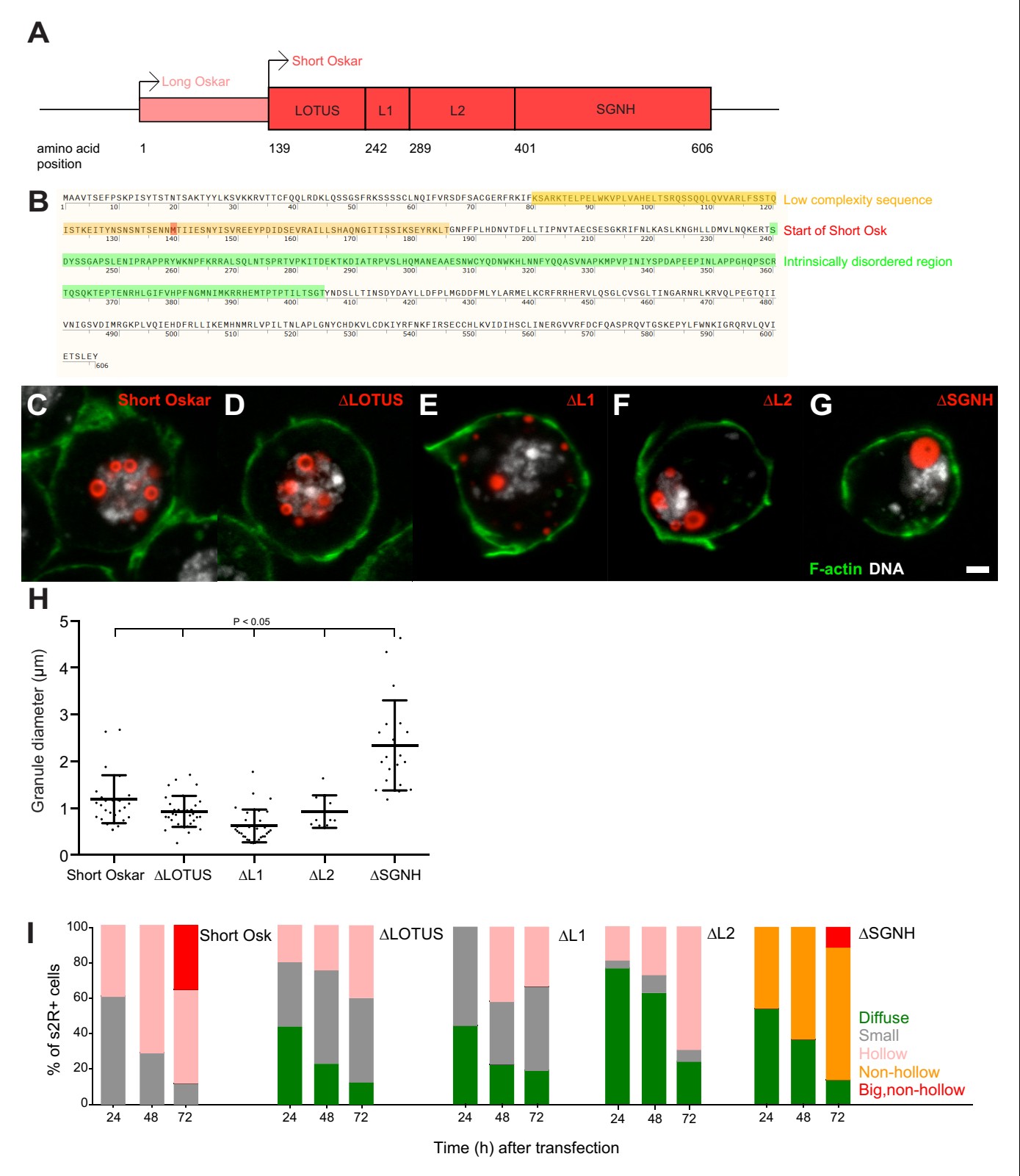

**Figure 5.** Multiple independent Oskar domains synergize to promote granule formation. (**A**) Schematic of Long and Short Osk. (**B**) Position of a predicted LC (orange) and IDR (green) in Oskar protein. The start aa of Short Osk is demarcated in red. (**C-G**) Expression of Short Osk:mCherry truncations (red) in S2R+ cells stained with DAPI for DNA (white) and phalloidin for F-actin (green). (**H**) Quantification of granule size in S2R+ cells transfected with Oskar or its truncations. Mean ± STDEV is plotted for each genotype. The mean diameter of ΔSGNH granules (2.33 μm) is larger (One-

*Figure 5 continued on next page*

Figure 5 continued

way ANOVA, p<0.05) than the diameter of granules formed by full length short Osk (1.19 μm), ΔLOTUS (0.93 μm), ΔL1 (0.62 μm) or ΔL2 (0.93 μm). (I) Condensation of full length Short mCherry:Osk and its truncations over time. Per time point, between 21 and 48 transfected cells per genotype were imaged and afterwards scored for the following phenotype: protein mostly present in a diffused form (green bar) or condensed into small (grey), hollow (pink), non-hollow (orange) and big, non-hollow (red) granules. Scale bar in C is 2 μm.

DOI: https://doi.org/10.7554/eLife.37949.022

The following figure supplement is available for figure 5:

**Figure supplement 1.** Multiple Short Osk domains synergize to promote granule formation.

DOI: https://doi.org/10.7554/eLife.37949.023

Oskar regions tested must act redundantly. In support, each domain appeared to affect the efficiency of granule formation (*Figure 5I*). Despite expressing similar amounts of protein (*Figure 5—figure supplement 1C*), cells with truncated versions contained significantly more diffusely distributed Short Osk (*Figure 5I*), which phase separated into larger and hollow granules later than full-

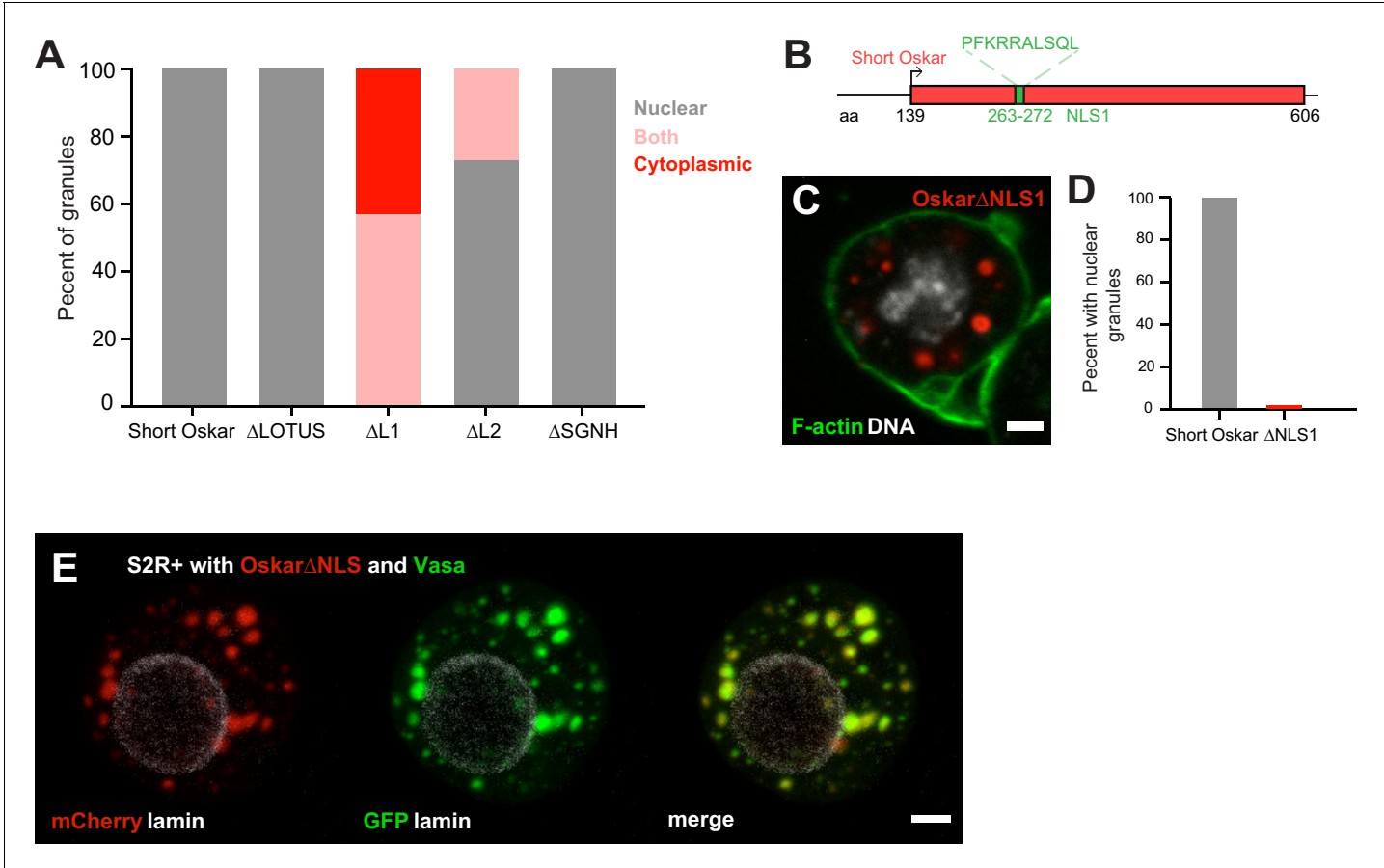

**Figure 6.** An NLS in Short Oskar controls its import into S2R + cell nuclei (see also *Figure 6—figure supplement 1*). (A) Cellular location of granules formed by transfection of full length short Osk or its truncations (n = 30 cells per genotype). (B) Schematic of NLS1 within L1. (C) Transfection of OskarΔNLS:mCherry in S2R+ cells (red) co-stained with DAPI (DNA, white) and phalloidin (green). (D) Percent of S2R+ cells with nuclear granules expressing full length mCherry:Osk or OskarΔNLS:mCherry (n = 30 cells per genotype). (E) Co-transfection of OskarΔNLS:mCherry (red) with Vasa:GFP (green) and co-stained with nuclear lamin (white). Scale bar in all images is 2 μm.

DOI: https://doi.org/10.7554/eLife.37949.024

The following figure supplement is available for figure 6:

**Figure supplement 1.** An NLS in Short Oskar controls its import into S2R + cell nuclei.

DOI: https://doi.org/10.7554/eLife.37949.025

length Short Osk (*Figure 5I*). Thus, multiple independent Osk protein domains synergize to promote efficient phase separation of Short Osk in S2R+ cells.

## An NLS in short Oskar controls its import into S2R + cell nuclei

Our structure-function experiments revealed that a signal responsible for the nuclear localization of Short Osk in S2R+ cells resided predominantly in Oskar's L1 region and to lesser extent in the L2 region (*Figure 5C–G*; *Figure 6A*, *Figure 5—figure supplement 1A,B*). Scanning the full-length aa sequence of Short Osk for nuclear localization sequences (NLS) identified two putative 10 aa long NLS motifs (*Figure 6—figure supplement 1A*), which fell within L1 (termed NLS1) and within L2 (termed NLS2) (*Figure 6B*, *Figure 6—figure supplement 1A*). To test the role of these predicted NLS sequences, we transfected S2R+ cells with constructs that carried a deletion of NLS1 (OskarΔNLS1) as well as a deletion of NLS1 and 2 (OskarΔNLS1 +2). OskarΔNLS1and OskarΔNLS1 +2 are unable to form nuclear granules but cytoplasmic granules formed, instead (*Figure 6C,D*, *Figure 6—figure supplement 1B,C*). Since OskarΔNLS1 completely prevented formation of nuclear germ granules (*Figure 6C,D*) and was able to partially support nuclear localization in the absence of NLS2 (*Figure 5I*, *Figure 6A*, *Figure 5—figure supplement 1A,B*), we conclude that in S2R+ cells NLS1 is the major contributor to the nuclear localization of Oskar granules. Importantly, deletion of NLS1 did not affect the ability of Short Osk to co-localize with Vasa in cytoplasmic granules (*Figure 6E*), indicating that the removal of NLS1 likely did not affect Oskar protein confirmation and left important aspects of its biological functions intact.

## Ablation of nuclear germ granules reduces cell divisions in PGCs

To address the function of nuclear germ granules in *Drosophila* germ line development, we deleted the NLS1 sequence from the endogenous *oskar* locus using CRISPR/Cas9. Using two guide RNAs targeting the NLS1 we recovered several in-frame deletions (*Figure 7—figure supplement 1A*) and selected alleles from separate injection experiments, termed CRISPR alleles A and B. To avoid phenotypic consequences due to off-target mutations, we tested the two deletion alleles in trans. We immunostained embryos derived from OskΔNLS/+ and OskΔNLS-A/OskΔNLS-B mothers for Oskar protein. In the control, OskΔNLS/+ embryos, nuclear germ granules were clearly visible both as small and hollow spheres within PGC nuclei (*Figure 7A*, white arrows, *Figure 7C,D*) while the OskΔNLS embryos failed to form such granules (*Figure 7B–D*). Small speckles visible inside PGC nuclei of the OskΔNLS embryos were most likely due to non-specific binding of Oskar antibody since we observed a similar signal in the neighboring somatic nuclei, which lack Oskar expression, (*Figure 7—figure supplement 1B,C*) and in the nuclei of Osk protein deficient embryos (*Figure 7—figure supplement 1D*) (*Vanzo and Ephrussi, 2002*).

We next addressed the biological role of nuclear germ granules. During embryogenesis, we observed nuclear germ granules as soon as syncytial nuclei reached the germ plasm. As PGCs formed, nuclear germ granules continued to increase in size and number (*Figure 2D,F,G*) such that within 22 min, the majority of PGCs contained nuclear germ granules (*Figure 2D*). Given their localization, we reasoned that nuclear germ granules likely play a role during or upon PGC formation but would not affect earlier stages of germ plasm assembly or function (see also below) (*Campos-Ortega and Hartenstein, 1985*; *Cinalli and Lehmann, 2013*; *Pae et al., 2017*; *Su et al., 1998*). Nuclear germ granules could facilitate PGC formation, PGC division or subsequent aspects of PGC development such as PGC survival, migration and maturation of PGCs into germline stem cells and gametes. We therefore counted the number of PGCs in fixed embryos laid by *OskΔNLS* mothers ('*OskΔNLS* embryos') and control *OskΔNLS/+* mothers ('*OskΔNLS/+* embryos') at NC 14 when PGC divisions have ceased(*Su et al., 1998*). We found that *OskΔNLS/+* embryos had an average of 36.9 ± 1.1 (n = 41) PGCs (*Figure 7E,G*) while *OskΔNLS* embryos only had 22.4 ± 1.2 (n = 42) (*Figure 7F,G*), a statistically significant 39.3% decrease (p<0.001) (*Figure 7G*). This reduction was not due to fewer nuclei migrating into the posterior pole as the number of PGC buds between the two genotypes was the same (*Figure 7H*). Rather, this result indicated that the reduction in PGC number occurred at a later step and was likely due to the reduced capability of *OskΔNLS* PGCs to cellularize, to divide or to be maintained.

To differentiate among these possibilities, we imaged PGC divisions in live embryos using light sheet microscopy. For precise developmental staging and analysis of mitosis, we introduced an RFP-

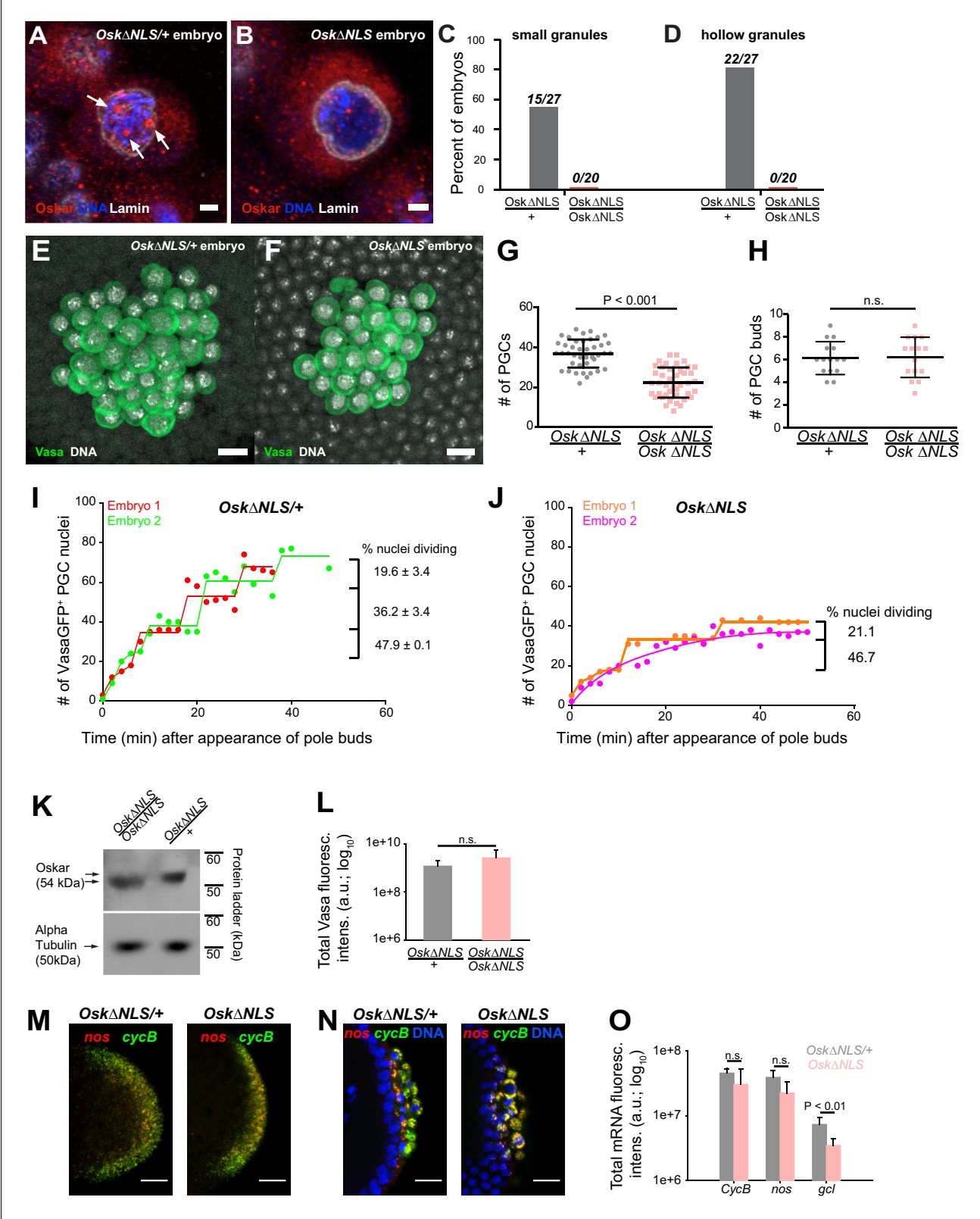

**Figure 7.** Ablation of nuclear germ granules reduces cell divisions in PGCs. (**A,B**) *Oskar∆NLS/+* PGCs (**A**) form nuclear germ granules (white arrows) while *Oskar∆NLS* PGCs do not (**B**). Granules were immunostained against Osk (red). DAPI-stained DNA is shown in blue and nuclear lamin is shown in white. (**C,D**) Number of embryos with small (**C**) and hollow (**D**) nuclear germ granules stained with α-Osk in *Oskar∆NLS* and *Oskar∆NLS/+* PGCs. In both the total number of embryos counted is written above the graph bars. (**E,F**) Cross-section of the posterior pole of an embryo with *Oskar∆NLS/+* (**A**) and

*Figure 7 continued on next page*

Figure 7 continued

OskarΔNLS (B) maternal genotype immunostained against Vasa (green) and DAPI (white). G Number of PGCs in 41 OskarΔNLS/+ and 42 OskarΔNLS embryos. Mean ± STDEV is plotted (statistical significance: unpaired t-test, p<0.001). (H) PGC buds in 15 OskarΔNLS/+ and 14 OskarΔNLS embryos. Mean ± STDEV is plotted (statistical significance: unpaired t-test). (I,J) Quantification of PGC divisions in live OskarΔNLS/+ (I) and OskarΔNLS (J) embryos. The emergence of the first PGC bud represented time 0 and the cellularization of somatic nuclei at NC 14 represented the end of the experiment. The number of Vasa:GFP associated HiS2Av:mRFP-stained nuclei was counted in three dimensions at every time point (red, green, orange and violet circles). Lines represent an estimated progression among successive cell divisions. Percent of dividing Vasa:GFP +nuclei is marked on each graph (*Figure 7—video 1* and *2*). Note, that in live observation, the total number of PGCs was higher than in fixed embryos (compare G with I and J). We attribute this to the fact that we counted all nuclei with even a small amount of Vasa:GFP associated, which may not have cellularized and become PGCs and thus would not have been counted as PGCs in fixed tissue. (K) Western blot of Oskar protein in OskarΔNLS and OskarΔNLS/+ embryos. α-tubulin was used for normalization control. L Total amount of Vasa germ plasm fluorescence in OskarΔNLS and OskarΔNLS/+ embryos marked with an antibody against Vasa. Mean total fluorescent levels ± STDEV of eight (OskarΔNLS/+) and 13 (OskarΔNLS) embryos per genotype are shown (statistical significance: two-tailed t-test). (M,N) Early embryos (M) and late embryos (N) hybridized with *nos* (red) and *CycB* (green) smFISH probes. DAPI-stained nuclei are shown in blue. O Quantification of localized mRNA levels hybridized with *CycB*, *nos* and *gcl* smFISH probes. Mean total fluorescent levels ± STDEV of four *CycB* and *nos*-stained embryos per genotype are shown and mean total fluorescent levels ± STDEV of four (OskarΔNLS/+) and five (OskarΔNLS) *gcl*-stained embryos are shown (statistical significance: two-tailed t-test). Scale bar in A,B is 2 μm and in E,F,M,N is 10 μm.

DOI: https://doi.org/10.7554/eLife.37949.026

The following video and figure supplement are available for figure 7:

**Figure supplement 1.** Ablation of nuclear germ granules reduces cell divisions in PGCs.

DOI: https://doi.org/10.7554/eLife.37949.027

**Figure 7—video 1.** PGC division in live OskarΔNLS/+ embryos.

DOI: https://doi.org/10.7554/eLife.37949.028

**Figure 7—video 2.** PGC division in live OskarΔNLS/OskarΔNLS embryos.

DOI: https://doi.org/10.7554/eLife.37949.029

tagged histone transgene (His2Av:mRFP), which marks the chromatin, and a Vasa:GFP transgene, which marks the germ plasm, into the OskΔNLS mutant background. As anticipated from our observations in S2R+ cells, Vasa:GFP accumulated in nuclear germ granules in embryos from OskΔNLS/+ control females but did not enter the nucleus in embryos from OskΔNLS mutant females (*Figure 3H, I*, *Figure 6E* and *Figure 7—figure supplement 1E,F*). We then collected embryos that had not yet formed PGC buds and acquired images of their posterior pole in three dimensions (3D) every two minutes in the RFP and GFP channels until NC 14, which can be clearly timed by the cellularization of the somatic nuclei. We counted the number of Vasa:GFP-associated nuclei in 3D at every time point for both genotypes where the appearance of the first pole bud represented time 0 min until NC 14 when in wild-type embryos the PGCs stop dividing (*Campos-Ortega and Hartenstein, 1985*; *Su et al., 1998*).

Our experiment revealed that in OskΔNLS/+ embryos PGCs divided synchronously relative to each other in the same embryo with some variation of cycle length between embryos (*Figure 7I*, red line shifted relative to the green one, *Figure 7—video 1*). In agreement with previous studies (*Su et al., 1998*), with each division, fewer and fewer PGCs divided (*Figure 7I*) indicating that as the embryo matured PGC divisions became less likely. In OskΔNLS embryos, however, synchronous divisions were not observed (*Figure 7J*, *Figure 7—video 2*). There were also fewer PGC divisions overall with a marked increase in the time between successive divisions (*Figure 7J*). As a result, by NC 14 there were 32.6 percent fewer PGCs in OskΔNLS embryos than in OskΔNLS/+ embryos (*Figure 7I,J*), in agreement with the results obtained with fixed samples (*Figure 7E–G*). After each division, PGCs of either genotype were retained at the posterior pole (*Figure 7—video 1* and *2*) indicating that newly divided PGCs did not die, were properly cellularized (i.e. formed) and maintained at the posterior pole. Thus, the reduction of PGCs in OskΔNLS embryos was due to the inability of mutant PGCs to divide properly.

Lastly, we asked whether the failure of mutant PGCs to divide was specifically due to the absence of nuclear Oskar or whether other aspects of Oskar's known functions were affected by the NLS mutation. Titration of Oskar protein affects the accumulation of germ plasm and of germ plasm-enriched mRNAs and, ultimately, the formation and specification of PGCs as well as the patterning of the embryonic anterior-posterior axis (*Lehmann, 2016*). Thus, the NLS mutation could have rendered Osk protein unstable or unable to accumulate adequately, which could have reduced the number of pole cells or affected embryonic development. However, we detected similar amounts of

Oskar protein in embryos of both genotypes (*Figure 7K*) as well as of germ plasm-enriched Vasa protein (*Figure 7L*). Furthermore, we observed no difference in the mRNA localization pattern (*Figure 7M,N*; note the characteristic 'RNA islands' around the dividing PGC nuclei) or in the levels of localized mRNAs such as *CycB* and *nos* (*Figure 7O*), respectively. In support, embryos laid by *OskΔNLS* mothers hatched like their heterozygous siblings suggesting that the amount of germ plasm-enriched *nos* mRNAs were similar between the two genotypes (*Figure 7—figure supplement 1H*)(*Gavis and Lehmann, 1994*). Importantly, PGCs in *OskΔNLS* embryos remained transcriptionally repressed (*Figure 7—figure supplement 1I*) indicating that mutant PGCs retained their germ line-specific nuclear character (*Hanyu-Nakamura et al., 2008*; *Martinho et al., 2004*). We did note that mutant embryos localized 52.6 percent less *gcl* mRNA (*Figure 7O*; *Figure 7—figure supplement 1G*), whose protein product controls the formation of PGCs. However, the initial formation of PGCs appeared normal in *OskΔNLS* embryos (*Figure 7—video 1* and *2*) and over-expression of *gcl* mRNA in *OskΔNLS* embryos did not rescue the reduced PGC number (*Figure 7—figure supplement 1J*). Therefore, the phenotype observed in *OskΔNLS* embryos is unlikely due to changes in the amount of germ plasm proteins or reduced enrichment of *gcl* mRNA but rather caused by defects in a previously unknown function of nuclear Oskar in the control of PGC nuclear divisions. Interestingly, the expression of Short Osk in S2R + cell did not affect their ability to divide (*Figure 7—figure supplement 1K–M*) indicating that this regulatory function of nuclear germ granules is restricted to PGCs (see Discussion).

## Discussion

First observed by EM over 50 years ago (*Mahowald, 1962*), we still know little about the biophysical and organizational properties of germ granules in any organism. Here we have used imaging methods and reconstitution experiments to analyze two types of embryonic germ granules in *Drosophila*: cytoplasmic germ granules and nuclear granules. While the function of cytoplasmic granules in germ cell formation and specification is well established, our study reveals a role for nuclear granules in the regulation of early PGC division. Quantitative fluorescent microscopy demonstrates that both granule types have biophysical properties of liquid condensates, where components rapidly diffuse between granules, as well as properties consistent with a hydrogel-like state, where proteins are stably associated within granules and do not exchange freely with their environment. Consistent with previous studies, our data show that short Oskar protein coordinates the assembly of both granule types but nuclear and cytoplasmic granules differ in their protein and RNA composition. Finally, we find that multiple domains contribute, apparently redundantly, to Oskar's ability to assemble germ granules and that an NLS within the Oskar protein coordinates granule assembly in the nucleus. Surprisingly, expression of Oskar in tissue culture cells reconstitutes aspects of nuclear granule assembly, thereby providing a system to study germ granules in a heterologous, non-germ line cellular system.

### Nuclear Oskar controls PGC division

Our studies identify a new function for Oskar, as a coordinator of PGC's cell division in the early *Drosophila* embryo. This function requires nuclear localization of the short isoform of Oskar. Nuclear germ granules are typically bigger, often appear hollow and fail to accumulate core germ granule constituents such as Aubergine and Tudor proteins and *nos*, *pgc* and *gcl* mRNAs consistent with these granules regulating different aspects of PGC development. PGC form four nuclear divisions prior to the cellularization of the somatic cells and thus become separated from the highly synchronized cell cycle control of the syncytial embryo. It is therefore attractive to hypothesize a role for nuclear granules in sequestering factors required for the coordinated division of early PGCs independently of the synchronous divisions of somatic nuclei. One model is that nuclear germ granules would increase the PGC number by sequestering negative regulators that would otherwise precociously inhibit PGC division. Such a role would not be unprecedented. In response to viral infections, cells form stress granules that sequester translational regulators to counteract the increased translational demands of a virus thereby suppressing its multiplication (*Reineke and Lloyd, 2013*). This regulator would be specific to germ cells and may explain why PGCs arrest at NC 14, while somatic cells continue to divide (*Su et al., 1998*). Furthermore, in support of a germ cell-specific regulator of cell division, cell divisions of somatic nuclei were unaffected by the NLS mutation and Short Oskar

nuclear germ granules did not affect the division of S2R+ cells devoid of core germ plasm constituents.

Alternatively, nuclear germ granules could enrich proteins that promote cell division thereby increasing their nuclear concentration and augment the likelihood of cells do divide. Indeed, several core components of the cell cycle machinery that promote DNA replication (Mcm3, Mcm5, and Dpa (Mcm4); *Figure 3—source data 1* [*Su et al., 1996*]) and progression through anaphase during mitosis (Apc7 and Cdc16; *Figure 3—source data 1* [*Harper et al., 2002*]) were enriched in the Short Oskar IP suggesting their sequestration in germ granules. Interestingly, *mcm5* and *mcm3* mRNAs are also posteriorly localized (*Lécuyer et al., 2007*; *Vourekas et al., 2016*) indicating that increased concentration of their gene products is important for germ line development. Indeed, cellularized PGCs divide slower than the somatic nuclei (*Su et al., 1998*), indicating that they are becoming limiting in factors that promote rapid cell division. In this model, PGCs in mutant embryos would 'run out' of limiting factors faster than their WT counterparts and stop dividing earlier. This model also explains why Short Oskar nuclear germ granules did not affect cycling of S2R+ cells as these are likely already dividing at a maximal rate. In this model, nuclear germ granules would only serve to accumulate and store the effector molecules that would promote division of PGCs, a function that is similar to cytoplasmic germ granules, which enrich and store mRNAs that code for effector proteins to promote the germ cell fate (*Lehmann, 2016*).

Lastly, it is also possible that diffuse nucleoplasmic Oskar itself directly regulates PGC division as a cell cycle regulator. In this scenario, the granules would act as a source of Oskar protein in the nucleus. Future work is needed to clarify the role of nuclear Oskar and nuclear germ granules in PGCs.

## Germ granules with similar organizational properties but distinct biological functions

Oskar nucleates both, cytoplasmic and nuclear germ granules and it is possible that Oskar and possibly Vasa proteins shuttle from the cytoplasm to the nucleus as PGCs form. FRAP, FLIP, photoconversion and experiments with aliphatic alcohols revealed that these two types of granules may use the same biophysical principles to achieve their distinct functions. For instance, their liquid properties could enhance biochemical reactions occurring within granules or provide a constant supply of diffusible protein to perform functions outside of granules. Conversely, the more stable conformation could ensure that their regulatory properties persisted throughout early embryonic development. Indeed, cytoplasmic germ granules begin forming and become functional in a transplantation assay during late oogenesis (*Illmensee et al., 1976*) and are maintained through early embryogenesis, a process that can last from 5 hr to many hours when fertilization is delayed (*Spradling, 1993*; *Su et al., 1998*). As a result, the enrichment of localized transcripts also persists, ensuring adequate levels of effector molecules to instruct germ cell fate. This could be achieved in part by making more stable, hydrogel-like organelles.

We found that multiple independent Short Oskar domains synergize to promote granule condensation, likely engaging both specific and non-specific protein interactions to de-mix, as proposed for other condensates (*Banani et al., 2016*; *Li et al., 2012*; *Lin et al., 2015*; *Protter et al., 2018*). However, this ability is lost in Long Oskar in cell lines and in embryos (*Markussen et al., 1995*; *Vanzo and Ephrussi, 2002*), which contains an additional 138 aa long N-terminus with a short LC domain (*Figure 5B*). This N-terminal extension is predicted to fold into three helices (data not shown), which could solubilize the protein and prevent its phase separation. Alternatively, the N-terminus could dictate a particular fold of Short Oskar in which the motifs promoting phase transition could become inaccessible. Interestingly, despite being coded by the same localized mRNA, the two isoforms are spatially segregated. Long Oskar is enriched cortically at the embryo's posterior and interacts with the actin network which is thought to stabilize germ granules (*Tanaka et al., 2011*), while Short Osk is concentrated within germ granules (*Rongo et al., 1997*; *Vanzo and Ephrussi, 2002*). Maintenance of germ granules is crucial for fertility and embryo development (*Arkov et al., 2006*). Thus, physical separation of Oskar isoforms is likely biologically relevant, perhaps to prevent granule dissolution by Long Osk should the two isoforms mix.

Despite years of research, it is largely unclear how the distinct functions of Oskar protein are accomplished with such high spatial and temporal precision. Our ability to understand the mechanisms by which Oskar realizes these roles has been limited, because the majority of *osk* mutations

prevent germ plasm assembly already during oogenesis and as a result preclude PGC formation and phase separation of nuclear germ granules. Several features of cytoplasmic and nuclear germ granules are recapitulated in nuclear germ granules formed in S2R+ cells including their cellular localization, morphology, composition and their biophysical properties. Thus, reconstitution of germ granules in S2R+ cells now provides an experimentally controllable system to study the distinct properties and functions of Oskar and its granules.

## Miscellaneous phase separated condensates - distinct or the same?

Germ granules are characteristic to germ cells of all species (*Eddy, 1975*) and many components, such as Vasa, Tud, Aub and granule mRNAs are conserved (*Voronina et al., 2011*), including in P granules of *C. elegans*. Despite morphologically, compositionally and functionally resembling P granules, our biophysical studies suggest that *Drosophila* germ granules behave more like aged yeast and mammalian stress granules by displaying both liquid-like and hydrogel-like properties (*Jain et al., 2016*; *Wheeler et al., 2016*). P granules have been proposed to largely behave as liquid droplets that form by liquid-liquid de-mixing (*Brangwynne, 2013*; *Brangwynne et al., 2009*). Interestingly, the more labile P granules are dispensable for germ line specification in *C. elegans* (*Gallo et al., 2010*), while germ granules in *Drosophila* have so far been inseparably associated with germ cell specification and early germ cell development. How these structural differences between *Drosophila* and *C. elegans* germ granules are realized in their function is unknown. Recent work on mammalian and yeast stress granules revealed that RNAs, the mini-chromosome maintenance (MCM) and RuvB-like (RVB) DNA helicase complexes stabilize these granules and promote their maturation from liquid droplets into hydrogels (*Burke et al., 2015*; *Jain et al., 2016*; *Zhang et al., 2015*). Some of these proteins including Mcm5, Mcm3, Dpa (MCM helicases) and pontin (RuvB-like helicase 1) associate with Short Osk by immunoprecipitation (*Hurd et al., 2016*) suggesting that they populate *Drosophila* germ granules, which could promote hydrogel-like conformation and provide rigidity to *Drosophila* germ granules.

RNA too can alter the biophysical properties of liquid droplets, decrease the viscosity of granules, their ability to exchange granule proteins with the environment and their propensity to fuse (*Zhang et al., 2015*). Indeed, we and others have noted that the cytoplasmic germ granules that are enriched with mRNAs (*Little et al., 2015*; *Trcek et al., 2015*) do not fuse (*Sinsimer et al., 2013*). We also could not observe fusion of nuclear germ granules in PGCs while fusion of nuclear germ granule in S2R+ cells was infrequent and slower than the fusion of P granules in *C. elegans* that form by LLPS (*Figure 4D*, *Figure 4—video 1* and *2*) (*Brangwynne et al., 2009*). These observations are consistent with the behavior of phase transitioned condensates (*Kato et al., 2012*; *Zhang et al., 2015*).

Many granules otherwise described as liquid droplets also display properties that by FRAP appear hydrogel-like (e.i. a fraction of their protein contents do not exchange with the granule environment, see *Figure 4—figure supplement 1C*). For example, only 16% of CAR-1 in P granules of *C.elegans* can exchange whereas other components appear to exchange more rapidly and readily (*Brangwynne et al., 2009*). Similarly, Dcp2 in P bodies of U2OS cells does not exchange while Dcp1a and b exchange to varying degrees (*Aizer et al., 2008*). It has been proposed that as granules age the polymerization of proteins in liquid-like droplets becomes enhanced and leads to an increased gelation within granules while the rest of the granule proteins retain liquid-like properties (*Lin et al., 2015*; *Wheeler et al., 2016*; *Xiang et al., 2015*). Thus, protein and RNA composition can influence polymerization behavior and may account for the varied biophysical properties even when the same component is analyzed.

Intriguing also is the spherical morphology and spatial organization of Oskar nuclear germ granules. By immunofluorescence and EM the larger granules have the appearance of a lighter, electron less dense core surrounded by a brighter, electron denser shell. We do not know whether this structure reflects any specific physical and functional properties. In the simplest scenario, Oskar could organize itself into a shell surrounding other protein or RNA components and therefore 'wet' the inner granule core. The core could itself be a phase separated condensate with miscible characteristics that would prevent mixing of the two protein phases(*Feric et al., 2016*). Indeed, several liquid phases with distinct biophysical and miscible properties form separate subcompartments within the frog nucleolus thought to facilitate sequential ribosomal RNA processing reactions (*Feric et al., 2016*). A similar structured organization has also been observed for MEG-3 and PGL-1 proteins in P

granules of *C. elegans* (*Wang et al., 2014*), as well as for mammalian stress granules composed of hydrogel-like core and liquid-like shel l(*Jain et al., 2016*; *Niewidok et al., 2018*).

Hollow condensates can form during a process called reentrant phase transition. Here, two oppositely charged counterions such as RNA and arginine-rich peptides mix such that at first the RNA stimulates phase separation of the peptide (*Banerjee et al., 2017*; *Milin and Deniz, 2018*). However, when the concentration of the RNA exceeds the concentration of the peptide, the RNA in the droplet de-mixes in the center of the granule and the protein droplets adopt a hollow appearance (*Banerjee et al., 2017*; *Milin and Deniz, 2018*). While smaller than their nuclear counterparts, cytoplasmic germ granules can also be hollow (*Arkov et al., 2006*). These are highly enriched with germ plasm-specific maternal mRNAs (*Little et al., 2015*; *Trcek et al., 2015*). mRNA demixing could induce the hollow morphology of these mRNA-bound germ granule. Intriguingly, the hollow appearance of nuclear granules was lost when Osk's RNA binding domain was deleted (*Figure 5G,I*). We do not know whether and if so which RNAs may occupy nuclear germ granules in PGCs as they are devoid of the known germ plasm enriched mRNAs (*Figure 2J*, *Figure 2—figure supplement 1F*). The hollow appearance of nuclear granules could be due to yet unknown RNA components, alternatively any oppositely charged counterion not just RNA can induce reentrant phase transition (*Banerjee et al., 2017*; *Milin and Deniz, 2018*).

The characteristic hollow morphology of nuclear germ granules is not as readily observed with cytoplasmic germ granules as it is with nuclear germ granules. However, these condensates display a unique RNA-protein organization where multiple localized mRNAs homotypically cluster at distinct positions within the granule center or at its periphery (*Trcek et al., 2015*). An uneven mRNA distribution is also observed within mammalian stress granules (*Jain et al., 2016*) and in cytoplasmic granules of the fungus *Ashbya gossypii* (*Zhang et al., 2015*).Thus, non-homogeneous organization appears to be a hallmark of diverse phase separated and phase transitioned condensates.

Despite intense research, many of the characteristics of phase separated condensates remain poorly understood; how is de-mixing spatially controlled within a cell, what proteins participate in this process, what is the biological relevance of their asymmetrical organization and what exact function do these granules perform. Experiments in a test tube provide a useful first tool to study the ability of proteins to demix into distinct granules (*Li et al., 2012*). However, they might not faithfully reflect how these proteins behave within a cellular and functional context (*Protter et al., 2018*) and may not allow to recapitulate the organization of the complete repertoire of protein and RNA components in a granule. Optimization of our cell culture approach could enable complete functional reconstitution of germ granules in somatic cells. Once established, this system would allow us to examine precisely how the organization of germ plasm components into granules specifies germline character.

## Materials and methods

### Key resources table

| Reagent type (species) or resource | Designation | Source or reference | Identifiers | Additional information |
|---|---|---|---|---|
| Antibody | Rabbit Poly(mono)clonal α-Oskar | A. Ephrussi | N/A | 1:1000 or 1:2500 (see Materials and methods) |
| Antibody | Rabbit polyclonal α-Vasa | R. Lehmann | N/A | 1:1000 |
| Antibody | Mouse monoclonal α-Lamin | Developmental Studies Hybridoma Bank | ADL84.12 | 1:1000 |
| Antibody | Alexa Fluor 488 Phalloidin | Molecular Probes | A12379 | 1:500 |
| Antibody | Rabbit polyclonal α-mCherry | Abcam | Ab167453 | 1:1000 |
| Antibody | Goat α-rabbit HRP | Abcam | Ab6721 | 1:2000 |
| Antibody | Goat α-rabbit horse HRP | Jackson Immuno Research | 111-035-003 | 1:5000 |
| Antibody | Mouse α -mouse HRP | Rockland | 18-8817-31 | 1:5000 |

*Continued on next page*

Continued

| Reagent type (species) or resource | Designation | Source or reference | Identifiers | Additional information |
|---|---|---|---|---|
| Antibody | Mouse monoclonal α-alpha tubulin | Sigma-Aldrich | T9026-2ML | 1:10000 |
| Antibody | Rabbit α-Aubergine | R. Lehmann | N/A | 1:1000 |
| Antibody | Rabbit α-Ser2 | Abcam | Ab5095 | 1:750 |
| Chemical | Vectashield Antifade Mounting medium | Vector Labs | Cat# H-1200 | |
| Chemical | Prolong Diamond Mounting medium | Molecular Probes | Cat# P36961 | |
| Chemical | Deionized formamide | Life Technologies | Cat# 4311320 | |
| Chemical | 10% Fetal Bovine Serum | Invitrogen | Cat#10082-139 | |
| Chemical | Penicillin-streptomycin | Fisher | Cat# 15140122 | |
| Chemical | Poly-L-lysine solution | Sigma | Cat# P4707-50ml | |
| Chemical | 20% Paraformaldehyde | Electron Microscopy Sciences | Cat#15713 | |
| Chemical | BSA | Sigma | Cat#A4503-100G | |
| Chemical | Restore Western Blot Stripping Buffer | Fisher Scientific | Cat# PI21059 | |
| Chemical | Triton X-100 | Sigma-Aldrich | Cat# T8787-250ML | |
| Chemical | 1,6 Hexanediol | Sigma-Aldrich | Cat#240117-50G | |
| Commercial assay | NuPAGE Novex 4-12% Bis-Tris Gel 1.0mm, 10 well | Invitrogen | Cat# NP0321BOX | |
| Commercial assay | Immun-Blot PVDF Membrane | Bio-Rad | Cat# 162-0174 | |
| Commercial assay | Imaging membranes | YSI | Standard | |
| Commercial assay | Effectine Transfection reagent | Qiagen | Cat# 301427 | |
| Commercial assay | Gibson Assembly Master Mix | New England BioLabs | Cat#E2611L | |
| Commercial assay | Nunc Lab-Tek II Chambered Coverglass | Thermo Fisher Scientific | Cat# 155409PK | |
| Cell line | D. melanogaster S2R+ | Drosophila Genomics Resource Center | FlyBase: FBtc0000150 | |
| Cell line | Human:HEK293T | ATCC | 293T (ATCC CRL3216TM) | |
| Organisms/Strains | Wild-type:w$^{-1118}$ | Bloomington (BDSC) | Stock #3605 | |
| Organisms/Strains | y,w; P[E GFP-vas w$^+$]$^{cyIII}$ | (*Trcek et al., 2015*) | N/A | |
| Organisms/Strains | *UAS–vasa–ko* | (*Cinalli and Lehmann, 2013*; *Trcek et al., 2015* ) | N/A | |
| Organisms/Strains | pFlyFos-Osk/CyO | (*Jambor et al., 2015*; *Trcek et al., 2015*) | N/A | |
| Organisms/Strains | UASp-GFP-Aub | (*Webster et al., 2015*) | N/A | |
| Organisms/Strains | GFP-Piwi | (*Le Thomas et al., 2013*) | N/A | |
| Organisms/Strains | GFP-Tud | R. Lehmann | N/A | |
| Organisms/Strains | *;;P{GAL4::VP16-nos.UTR}/Tm3Ser* | R. Lehmann | N/A | |
| Organisms/Strains | w[*]; P{w[+mC]=His2Av-mRFP1}II.2 | Bloomington (BDSC) | Stock #23651 | |
| Organisms/Strains | *w;;OskΔNLS-alleleA/TM3* | This study | N/A | See Materials and methods |
| Organisms/Strains | *w;;OskΔNLS-alleleB/TM3* | This study | N/A | See Materials and methods |
| Organisms/Strains | *w;His2AAv-mRFP/cyo;OskΔNLS-alleleA,Vasa:GFP/TM3* | This study | N/A | See Materials and methods |

*Continued on next page*

*Continued*

| Reagent type (species) or resource | Designation | Source or reference | Identifiers | Additional information |
|---|---|---|---|---|
| Organisms/Strains | *w; matα-gal4;PrDr/TM3* | (*Pae et al., 2017*) | N/A | |
| Organisms/Strains | *Df(3R)Pxt103/TM3 Ser Sb* | (*Lehmann and Nüsslein-Volhard, 1991*) | N/A | |
| Organisms/Strains | *osk54 st ry ss/TM3Ser* | (*Lehmann and Nüsslein-Volhard, 1991*) | N/A | |
| Organisms/Strains | *w; P(EPgy2)gcl^EY09611/Cyo; nos Gal4VP16 (w+)/TM3* | (*Cinalli and Lehmann, 2013*) | N/A | |
| Organisms/Strains | *FT553: zuIs247 [nmy-2::PGL-1-GFP line O05-2]; unc-119(ed3) III; ltIs44 V [pie-1p-mCherry:: PH(PLC1delta1)]* | J. Nance | N/A | |
| Oligonucleotides | Osk nls gRNA-A | This study | N/A | See Materials and methods |
| Oligonucleotides | Osk nls gRNA-B | This study | N/A | See Materials and methods |
| Oligonucleotides | ssODN for gcl-NLS-A | This study | N/A | See Materials and methods |
| Oligonucleotides | ssODN for gcl-NLS-B | This study | N/A | See Materials and methods |
| Oligonucleotides | Calca Fluor 590-conjugated *nanos* smFISH probes | (*Trcek et al., 2015*) | N/A | |
| Oligonucleotides | Quasar 670-conjugated *gcl* smFISH probes | (*Trcek et al., 2017*) | N/A | |
| Oligonucleotides | Quasar 670-conjugated *cycB* smFISH probes (5′→3′): cgttttgttgttgcctccat, tatgccgcgattctgcaaat, tctttctgtgccgcatcctt, tctgtgagcttgagatcctt, tccacccgagctttggcatt, agtggctgtttcttccagtg, ttgccattgccattggtgct, aattcgaacgcaaaaacgcc, tacagtggtcttggtcggaa, tgactgtaactttagtgggt, ttcacgttctcggaagaact, ttgctgtcctcgcgctttaa, agtttggtcagcgacttctt, attcctgaaactcccatcac, aaacagctactggttcccgt, ttcttggtctctgcctcttt, cttcttggtttctggcagtt, cctttttcacttccagtgag, tgcactgttgcccctaataa, atcgtagatgtggtcgtagt, ttgctggaaagggacatggt, atcaatgtcctcgattccag, accaggttctccttgtcatt, cgtttacatattcggagacc, cctgatacaagtagtcgtag, atccttgtgaatgggctgct, atcttgtgggacacctcctt, ttgatccaatcgatcagcac, atggaactgcaggtggactt, tagcgatcaatgatagccac, tttggtgtccttgaccacct, actcccaccaattgcaagta, acttggtggctatgaagagt, aagacgaaatctccgattgc, agtgtaggtgtcgtccgtga, agatttgcagctccatctgt, agattacagtcgatggcctt, tagcgtcgaaggaagtgaat, ttggacatcgtatggtgctc, ggaagctaactcgatgaagt, taagtggccatttcgtagtc, aacagtgaggcagctgcaat, tttccattgagcaagtgcag, acggtcgttgaatcctgtac, atcgcgagtagaaggtcaga, ttcgcgatcagccgggtaat, ttgtagatggccttcagctt, atcttctggaacttgctgcc | This study | N/A | See Materials and methods |
| Recombinant DNA | pActin5C | *Drosophila* Genomics Resource Center | Cat#1112 | |
| Recombinant DNA | pVERMILION22 | (*Ni et al., 2011*) | N/A | |
| Recombinant DNA | pEF1/V5-His A | ThermoFisher Scientific | V92020 | |
| Recombinant DNA | pDendra2 | Addgene | Cat#75283 | |
| Software and Algorithms | ImageJ | NIH | https://imagej.nih.gov/ij/ | |

*Continued on next page*

*Continued*

| Reagent type (species) or resource | Designation | Source or reference | Identifiers | Additional information |
|---|---|---|---|---|
| Software and Algorithms | NLS finder | cNLS Mapper | http://nls-mapper. iab.keio.ac.jp/ cgi-bin/NLS_ Mapper_form.cgi | |
| Software and Algorithms | IDR and LC prediction | SEG IUPRED JPRED | http://mendel.imp. ac.at/METHODS/ seg.server.html http://iupred. enzim.hu/ http://www.compbio. dundee.ac.uk/jpred/ | |
| Software and Algorithms | EasyFRAP | (*Rapsomaniki et al., 2012*) | N/A | |
| Software and Algorithms | Huygens deconvolution software | Scientific Volume Imaging | N/A | |
| Software and Algorithms | Prism 7 | Graphpad | N/A | |
| Software and Algorithms | SigmaPlot | SigmaPlot (Systat Software, Inc) | N/A | |
| Software and Algorithms | CRISPR design for guide RNAs | MIT | http://crispr.mit.edu/ | |
| Software and Algorithms | CRISPR design for guide RNAs | ZiFit | http://zifit.partners. org/ZiFiT/ | |
| Software and Algorithms | DAVID GO term analysis | The Database for Annotation, Visualization andIntegrated Discovery (DAVID) version 6.7. | https://david. ncifcrf.gov/ | |

## *Drosophila melanogaster* and *Caenorhabditis elegans*

Flies were maintained on cornmeal molasses/yeast medium at 25°C using standard procedures. The following fly lines were used: Wild-type:w$^{-1118}$(Bloomington (BDSC); Stock #3605), y,w; P[E GFP-vas w$^{+}$]$^{cyIII}$ (*Trcek et al., 2015*), UAS–vasa–KO (*Cinalli and Lehmann, 2013*; *Trcek et al., 2015*), pFlyFos-Osk (*Jambor et al., 2015*; *Trcek et al., 2015*), UASp-GFP-Aub (*Webster et al., 2015*), GFP-Piwi (*Le Thomas et al., 2013*), GFP-Tud (*Zheng et al., 2016*), ;;P{GAL4::VP16-nos.UTR}/TM3Ser (*Pae et al., 2017*), w[*]; P{w[+mC]=His2Av-mRFP1}II.2 (Bloomington (BDSC); Stock #23651), w; matα-gal1;PrDr/TM3 (*Pae et al., 2017*), w; P(EPgy2)gcl$^{EY09611}$/CyO; nosGal4VP16 (w+)/TM3 (*Cinalli and Lehmann, 2013*), Df(3R)Pxt103/TM3 Ser Sb and osk54 st ry ss/TM3Ser (*Lehmann and Nüsslein-Volhard, 1991*).

To generate *OskarΔNLS* using CRISPR/Cas9, two separate injections were performed into *Drosophila* embryos transgenically expressing Cas9 under the Vasa promoter (stock BDSC#55821). Each injection contained two plasmids to express CRISPR guide RNAs targeting the Oskar NLS and a single-stranded DNA oligo to promote homologous recombination between the two cut sites. After the guide RNAs were identified and checked for genome-wide uniqueness (http://crispr.mit.edu/ and http://zifit.partners.org/ZiFiT/), they were cloned into the pU6-BbsI-chiRNA plasmid (http://fly-crispr.molbio.wisc.edu/protocols/gRNA). Flies were screened by PCR and sequenced to isolate two germline integrations of in-frame deletions within the Oskar NLS, w;; OskΔNLS-alleleA/TM3 and w;; OskΔNLS-alleleB/TM3.

Worms (FT553: zuls247 [nmy-2::PGL-1-GFP line O05-2]; unc-119(ed3) III; ltls44 V [pie-1p-mCherry::PH(PLC1delta1)]; gift from J. Nance, NYU) were maintained on nematode growth medium seeded with OP50 *E.coli* strain at room temperature (RT) using standard procedures.

## Tissue culture

*Drosophila* S2R+ cells (DGRC; FBtc0000150) were maintained at 25°C and 5% $CO_2$ in Schneider's medium (Gibco) containing 10% Fetal Bovine Serum and 1% Penicillin-streptomycin. Effectene was used to transfect 200 ng of each plasmid (unless specified otherwise). Transfected cells were incubated for at least 24 before use. Human HEK293 cells (293T (ATCC CRL3216TM) were purchased from ATCC in 2006 and low passage number vials were cryopreserved. The cells are mycoplasma negative. ATCC authenticated cells by isoenzymology tests to confirm species and morphology.

Once thawed, cells were maintained at 37°C in Dulbecco's modified Eagle's medium supplement L-Glu and containing 10% Fetal Bovine Serum and 1% Penicillin-streptomycin. Effectene was used to transfect 200 ng of plasmid. Transfected cells were incubated for at least 24 hr before use.

## Preparing embryos and S2R+ cells for live imaging

Embryos were dechorionated as previously described (*Trcek et al., 2017*) and afterwards affixed onto the heptane glue-coated coverslip, covered with halocarbon oil and placed into an imaging chamber with a central hole covered with imaging membrane to allow gas exchange.

24 hr after transfection, S2R+ cells, resuspended in growth medium, were affixed onto the Poly-L-lysine-coated Lab-Tek dishes. All videos were recorded at RT with a laser scanning microscope.

## Immunostaining embryos

Dechorionated embryos (*Trcek et al., 2017*) were fixed for 45 min at RT, devitellinized by hand and stained overnight at 4°C with rabbit α-Oskar (kind gift of Dr. Anne Ephrussi; 1:1000), mouse monoclonal α-lamin (Developmental Studies Hybridoma Bank; ADL84.12;1:1000), rabbit polyclonal α-Vasa ([*Trcek et al., 2015*]; 1:1000), Alexa Fluor488 phalloidin (Molecular Probes; A12379; 1:500) or Rabbit α-Ser2 (Abcam; Ab5095; 1:750) 1°antibodies.

## Immunostaining S2R+ cells

Transfected cells were attached to Poly-L-lysine coated coverslips and fixed with 4% paraformaldehyde in 1XPBS for 10 min. Afterwards, they were permeabilized in 1X PBS containing 0.1% Triton X-100 for 15 min and stained overnight at 4°C with mouse monoclonal α-lamin (1:1000) or Alexa Fluor488 phalloidin (1:500) 1°antibodies.

## Immunoblotting embryo extracts

Protein was extracted from 0 to 40 min old embryos, run on a SDS-polyacrylamide gradient gel and then transferred to a PVDF membrane. The membrane was probed with α-Oskar (1:2500) diluted in PBST +2% skim milk powder and then with goat α-rabbit horse HRP (Jackson ImmunoResearch; 111-035-003;1:5000). Protein bands were detected using chemiluminescence. The membrane was then stripped using Restore Western Blot Stripping Buffer and re-probed with mouse monoclonal α-alpha tubulin (Sigma-Aldrich; T9026-2ML; 1:10000) and α mouse α -mouse HRP (Rockland; 18-8817-31; 1:5000) and detected by chemiluminescence.

## Immunoblotting S2R + cell extracts

S2R+ cells were co-transfected with 140 ng Short Osk:mCherry or its truncations and 140 ng Aub: GFP plasmids. After 72 hr, cells were washed and resuspended in 1XPBS. Cell suspension was mixed with equal amount of Laemmli sample buffer, boiled at 95°C for 5 min and run on a SDS-polyacrylamide gradient gel and then transferred to a PVDF membrane. The membrane was blotted with rabbit polyclonal α-mCherry (Abcam; Ab167453; 1:1000 dilution) and then goat α-rabbit HRP (Abcam, Ab6721; 1: 2000 dilution). Membrane was imaged with chemiluminescence. The membrane was then stripped with Restore Western Blot Stripping Buffer and re-probed with rabbit α-Aubergine ([*Trcek et al., 2015*]; 1:1000 dilution) and goat α-rabbit HRP (1: 2000 dilution) and imaged with chemiluminescence. Integrated intensities of protein bands were quantified with ImageJ. Intensity of each Oskar protein band was then normalized by the intensity of the corresponding Aub:GFP band to determine its expression level.

## smRNA FISH

NC 1–14 embryos were fixed and afterwards commercially available Stellaris RNA smFISH probes were used to label *nos* and *gcl* as described previously (*Trcek et al., 2015*; *Trcek et al., 2017*). To detect *CycB*, the following Quasar670-conjugated Stellaris smFISH probes were used: cgttttgttgttgcctccat, tatgccgcgattctgcaaat, tctttctgtgccgcatcctt, tctgtgagcttgagatcctt, tccaccc-gagctttggcatt, agtggctgtttcttccagtg, ttgccattgccattggtgct, aattcgaacgcaaaaacgcc, tacagtggtcttggtcggaa, tgactgtaactttagtgggt, ttcacgttctcggaagaact, ttgctgtcctcgcgctttaa, agtttggt-cagcgacttctt, attcctgaaactcccatcac, aaacagctactggttcccgt, ttcttggtctctgcctcttt, cttcttggtttctggcagtt, ccttttttcacttccagtgag, tgcactgttgcccctaataa, atcgtagatgtggtcgtagt, ttgctggaaagggacatggt,

atcaatgtcctcgattccag, accaggttctccttgtcatt, cgtttacatattcggagacc, cctgatacaagtagtcgtag, atccttgt-gaatgggctgct, atcttgtgggacacctcctt, ttgatccaatcgatcagcac, atggaactgcaggtggactt, tagcgatcaatga-tagccac, tttggtgtccttgaccacct, actcccaccaattgcaagta, acttggtggctatgaagagt, aagacgaaatctccgattgc, agtgtaggtgtcgtccgtga, agatttgcagctccatctgt, agattacagtcgatggcctt, tagcgtcgaaggaagtgaat, ttgga-catcgtatggtgctc, ggaagctaactcgatgaagt, taagtggccatttcgtagtc, aacagtgaggcagctgcaat, tttccattgag-caagtgcag, acggtcgttgaatcctgtac, atcgcgagtagaaggtcaga, ttcgcgatcagccgggtaat, ttgtagatggccttcagctt, atcttctggaacttgctgcc.

## Microscopy

Majority of samples were imaged with a Zeiss LSM780, AxioOberver inverted laser scanning confocal microscope, equipped with an argon, an HeNe 633 laser and a DPSS 561–10 laser, a Plan-Apo40X/1.4 Oil DIC and EC Plan-Neofluar 10X/0.30 objectives. To quantify Vasa levels in the embryos immu-nostained with α-Vasa and to record the dynamics of granule formation by full length Oskar or its truncations, a Zeiss AxioObserver.Z1 Widefield Epifluorescence microscope equipped with the Plan-Apochromat 63x/1.40 Oil DIC M27 objective, the FL Filter Set 49 DAPI, FL Filter Set 38 HE GFP, FL Filter Set 43 HE Cy3 filter sets and the Cond LD 0.55 hr/DIC/Ph 6x Mot light source and the Axiocam 503 Mono camera was used. Images were acquired in 3D and afterwards deconvolved using Huy-gens deconvolution software (Scientific Volume Imaging). To record PGC divisions in live embryos, a Zeiss Lightsheet Z.1 microscope equipped with water immersion Plan-Apochromat 20x/1.0 UV-VIS (Serial No. 4909000088) detection objective, 488 nm and 561 nm lasers, and band pass 505 nm-545nm and long pass 585 nm filters were used. The ZEN 2014 SP1 (black edition) version 9,2 soft-ware was used to process the images.

## Correlative light and electron microscopy

Transfected S2R+ cells were prepared for electron microscopy by the NYULH DART Microcopy Core Laboratory. Cells were cultured on a 35 mm No. 1.5 gridded glass bottom dish (MatTek Cor-poration). After cells were imaged under the Zeiss Axio Observer epifluorescence and crossed polar-ization microscope to obtain the fluorescence and phase images, they were then fixed in 2.5% glutaraldehyde and 2% paraformaldehyde in 0.1M Hepes buffer (pH 7.2) at room temperature for one hour, and afterwards continued to be fixed at 4°C overnight. The cells were then post-fixed with 1% osmium tetroxide for 1 hr and stained with 1% uranyl acetate in ddH$_2$O at 4°C overnight. Dehy-dration was carried out at room temperature using serial ethanol solutions. The cells were *en face* embedded with Araldite 502 (Electron Microscopy Sciences, Hatfield, PA), and polymerized at 60°C for 48 hr. The sample block was removed by immersing the whole dish in liquid nitrogen, and area of interest then trimmed under stereomicroscope. The grid pattern imprinted in the resin served as the landmark to correlate optical and fluorescence image. 70nm serial ultrathin sections were cut using Leica UC6 ultramicrotome (Leica Microsystems Inc.), collected on formvar coated slot copper grids and stained with uranyl acetate and lead citrate by standard methods. The surface 1–2 sections were cut at 100 nm containing marker grid pattern and afterwards recognized under transmission electron microscope (Philips CM-12, Thermo Fisher, Eindhoven, The Netherlands), and serial sections were captured with a Gatan (4k × 2.7 k) digital camera (Gatan, Inc., Pleasanton, CA).

## Quantification of granule protein enrichment

Two different measures were used to quantify the fold enrichment of proteins within granules rela-tive to granule surroundings; by concentration fold enrichment (*Figures 1B*, *2H* and *4A*) and by quantifying the percent of non-granular nucleoplasmic protein in PGC and S2R + cell nuclei (*Fig-ure 2—figure supplement 1D*, *Figure 4—figure supplement 1A*). To quantify the concentration fold enrichment of Osk:GFP in cytoplasmic germ granules, embryos expressing the Osk:GFP trans-gene were first imaged in 3D with a laser scanning confocal microscope. Afterwards, 23 granules were segmented in ImageJ and total fluorescence intensity normalized by the granule area. Only the equatorial plane (where the granule is most in focus) was analyzed. The same analysis was then pre-formed using ROIs within the intergranular space and the somatic cytoplasm (*Figure 1—figure sup-plement 1B*). Means were compared to determine the fold difference in the fluorescent intensity of Osk:GFP located in the granules versus the intergranular space (*Figure 1B*). The same approach was employed to quantify the fold enrichment of granular Osk and Vasa in *Figure 2H*, and *Figure 4A*.

To quantify the total amount of non-granular nucleoplasmic protein in PGCs and S2R+ cells, fixed embryos and S2R+ cells were imaged as described above. Afterwards, individual PGC or S2R + cell nuclei were segmented using the DAPI stain and total Oskar or Vasa nuclear fluorescence intensity determined using the 3D object counter plugin in ImageJ (*Bolte and Cordelières, 2006*). Using the same plugin, individual nuclear granules were segmented and their total fluorescence intensity determined. After subtraction, the percent of non-granular nucleoplasmic Oskar or Vasa protein was determined (% nucleoplasmic content).

## Quantification of granule sizes and protein abundance

Using ImageJ, lines were drawn through several granules imaged in their equatorial plane and their linear fluorescent profiles extracted using a Plot Profile plugin. Pixels in these profiles where the fluorescence intensity started to increase relative to background represented the edges of the granule (i.e. size in nm), while the summed fluorescence intensity contained within these pixels represented the total fluorescence intensity (i.e. protein abundance) of an individual granule.

## Fluorescence Recovery after Photobleaching (FRAP)

Live embryos and S2R+ cells expressing Osk:GFP, Vasa:GFP and Short mCherry:Osk transgenes were prepared as described above. To FRAP Osk:GFP or Vasa:GFP in the germ plasm, an ROI (x = y three μm) located in the middle of germ plasm was bleached using a single strong pulse of 488 laser illumination and afterwards an image of the germ plasm acquired every second to record fluorescence recovery. Initial five images were acquired to establish the levels of pre-bleach fluorescence. To FRAP Osk:GFP or Vasa:GFP in nuclear germ granules in PGCs, a single nuclear granule was bleached per embryo and imaging carried out as described above. To FRAP Short Osk:mCherry in nuclear germ granules in S2R+ cells, a single nuclear granule was bleached per cell. In this case, a single strong pulse of 561 laser illumination was used for bleaching. Imaging was performed using 561 laser illumination as described above.

Fluorescence fluctuations in the bleached ROIs were extracted in ImageJ. Afterwards, using easyFRAP (*Rapsomaniki et al., 2012*), a full-scale normalization procedure was used to normalize recovery curves (*Brangwynne et al., 2009*; *Rapsomaniki et al., 2012*). We used the full scale normalization because it corrects for differences in bleaching depth among different experiments. As such, all normalized recovery curves started from 0 value and could be easily compared with each other (*Rapsomaniki et al., 2012*). Normalized curves were then averaged and fit to a single term exponential equation ($f(t)=a*(1-exp(-b*t))$) in Sigmaplot, where $a$ represented the percent mobile fraction and $b$ represented the rate constant of fluorescence recovery. Afterwards, $b$ was used to calculate the half time to full fluorescence recovery $t_{1/2}$ (s) using the equation $b = ln(2)/t_{1/2}$ (*Brangwynne et al., 2009*; *Rapsomaniki et al., 2012*).

To record fluorescence recovery of a partially bleached nuclear mCherry:Osk granule in S2R + cells, approximately half of the granule was bleached and afterwards imaged as described above. The fluorescence recovery was recorded, recovery curves normalized and analyzed as described above.

## Fluorescence Loss in Photobleaching (FLIP)

Live NC 1–5 embryos expressing transgenic Osk:GFP or Vasa:GFP were prepared as described above. To record FLIP, a single ROI (x = y three μm, region A) located in the middle of the germ plasm was continuously bleached using a single strong pulse of 488 laser illumination while concurrently acquiring an image of germ plasm every second for the duration of four minutes. Initial five images were acquired to establish the pre-bleach fluorescence levels. Fluorescence loss in the bleached region A as well as in the neighboring regions B, C, D (x = y three μm) was extracted in ImageJ. These values were then normalized to the initial value (time 0 s) and expressed as a percent (% fluorescence remaining). To determine the rates of fluorescence loss due to unintended photobleaching, an ROI (x = y three μm) located outside of the embryo was continuously bleached and % fluorescence remaining for regions A, B, C and D located in the germ plasm determined as described above. No additional normalization steps as described for FRAP experiments were performed. For both transgenes, unintentional bleaching reduced the level of fluorescence by five to 24% (*Figure 1—figure supplement 1Di,Fi*), intentional bleaching within region A by 95%

(*Figure 1Dii*, *Figure 1—figure supplement 1Ei*), and redistribution of Oskar and Vasa from ROI B into neighboring ROIs by 45% to 59%, respectively (*Figure 1Dii*, *Figure 1—figure supplement 1Ei*). Thus, compared to unintentional bleaching, redistribution of granule protein into neighboring ROIs accounted for redistribution of minimally 20% to 30% of the protein.

## Quantifying granule numbers and phenotypes in S2R+ cells

Cells expressing Short mCherry:Osk or its truncations were fixed 24, 48 and 72 hr after transfection. At each time point, expressing cells were binned into categories depending on the size and appearance of the granules they formed (small, hollow, non-hollow, big and non-hollow). If cells expressed predominantly diffused protein and formed only a few small granules, then the cells were categorized as belonging to a 'diffused' category.

## Live imaging of granules in S2R+ cells

To detect granule fusion, transfected cells were rapidly imaged in a single plane or in 3D every minute. To detect granule dissolution, 4% or 15% of 1,6 Hexanediol, 2,5-Hexandiol, 1,5-Pentandiol or 1,4-Butandiol (*Kroschwald et al., 2015*; *Lin et al., 2016*; *Patel et al., 2007*) resuspended in S2R + cells growth medium was added to the cells mounted in Lab-Tek dishes. Cells were imaged 20 min later. To detect exchange of Short Dendra2:Osk among granules, green fluorescence of three granules within the same nucleus was photoconverted to red fluorescence using a strong pulse of the 488 laser illumination. Images before and after photoconversion were acquired every second using a 561 laser illumination. All videos were recorded with a laser scanning confocal microscope

## Isolation of short Osk:Dendra2 granules

24 hr after transfection, S2R+ cells expressing Short Osk:Dendra2 were harvested with centrifugation at 1000 g for 2 min. Cells were then washed once with phosphate-buffered saline (PBS) and lysed in hypotonic buffer (10 mM Tris-HCl pH7.5, 10 mM KCl, 1.5 mM MgCl2) supplemented with 0.2% IGE-PAL CA-630 (Sigma-Aldrich) for 5 min. Lysate was centrifuged at 220 g for 5 min. Pelleted nuclei were resuspended in hypotonic buffer and sonicated for 6 × 10 s bursts (with 10 s intervals between each burst) with Bioruptor 300 sonication system in high power output. Sonicated sample was layered over sucrose buffer (0.88M sucrose, 0.5 mM MgCl2) and centrifuged at 3000 g for 10 min. Pellet containing nuclear granules was resuspended in hypotonic buffer for further experiments.

## Predicting LC and IDR domains

Sequence analyses to predict LC and IDR regions of Oskar were carried out with a combination of SEG (http://mendel.imp.ac.at/METHODS/seg.server.html), IUPRED (http://iupred.enzim.hu/) and JPRED (http://www.compbio.dundee.ac.uk/jpred/). Standard values (pre-assigned in the programs) were used as the search parameters.

## Counting PGCs and PGC buds

Fixed NC nine embryos (to count PGC buds) and NC 14 embryos (to count PGCs) were immunostained with α-Vasa. Afterwards, embryo's posterior was cut with a razor blade, mounted on a plane parallel to the slide (PGCs facing up) (*Slaidina and Lehmann, 2017*) and imaged in 3D.

## Quantification of Vasa in germ plasm

Early embryos (NC 1 to 8) were fixed and immunostained using α-Vasa. Posterior ends of embryos were then imaged in 3D and total Vasa fluorescence intensity using a 3D object counter plugin in ImageJ (*Bolte and Cordelières, 2006*).

## Quantifying smFISH-hybridized embryos

Early embryos (NC 1 to 5) were fixed and hybridized with smFISH probes targeting *CycB*, *nos* or *gcl* mRNAs as described previously (*Trcek et al., 2017*). Posterior ends of embryos were imaged using a laser scanning microscope in 3D and total fluorescence intensity of smFISH-hybridized mRNAs quantified using a 3D object counter plugin in ImageJ (*Bolte and Cordelières, 2006*).

## NLS prediction

Oskar NLSs were predicted using cNLS Mapper (http://nls-mapper.iab.keio.ac.jp/cgi-bin/NLS_Mapper_form.cgi).

## Cell sorting and cell cycle analysis

Live transfected S2R+ cells were washed once in 1XPBS and resuspended in 1XPBS to the concentration of 10–20 million cells/ml. Afterwards, they were sorted into mCherry positive and mCherry negative cells using the MoFlo cell sorter (Beckman Coulter). Each cell population was then fixed with 80% ethanol on ice and DAPI stained to label the DNA. Using the LSRII cell cycle analyzer (BD), DAPI intensity of single cells was determined while excluding cell clumps and aggregates. The distribution of DAPI intensities (*Figure 7—figure supplement 1L,M*, black line) was fitted using ModFit (Verity Software House) (*Figure 7—figure supplement 1L,M*, purple line) to extract individual cell cycle components and determine the percent of cells belonging to G1, S and G2/M phases of the cell cycle.

## Egg laying and egg hatching rates

For each genotype, seven up to two-day old virgin females were mated with four WT males for four days and afterwards placed in an egg collection cage covered with a three-cm round apple juice plate containing a dollop of fresh yeast paste (*Trcek et al., 2017*). Cages were then placed into a 25°C humidity-controlled incubator for 24 hr. Flies were allowed to lay eggs on fresh plates for 2 hr, plates removed and laid eggs counted. Plates were then returned into the incubator and the number of hatched eggs counted 48 hr later.

## Statistical analysis, statistical reporting and sample size estimation

Statistical analysis, information about the sample size and technical replicates for each experiment are provided in the figure legends. No explicit power analysis was used to estimate the sample size used for each experiment.

## Acknowledgements

We would like to thank the Lehmann lab members for helpful suggestions, and Drs. Emily Dawson, Toby Lieber and Steve McKnight for critical reading of the manuscript, Dr. Jeremy Nance for sharing the PGL-1:GFP worm line, Mamta Tahiliani for sharing Hek293 cells and Sophia He for her help with the western blots. *Drosophila* stocks and plasmids were provided to us by the *Drosophila* Genomics Resource Center supported by NIH grant 2P40OD010949. Cell sorting/flow cytometry technologies and EM Microscopy were provided by NYU Langone's Cytometry and Cell Sorting Laboratory and Microscopy Core, respectively, which are supported in part by grant P30CA016087 from the National Institutes of Health/National Cancer Institute. This work was supported by the NICHD K99HD088675 grant awarded to TT. RL is an HHMI investigator.

## Additional information

### Funding

| Funder | Grant reference number | Author |
| --- | --- | --- |
| Howard Hughes Medical Institute | | Ruth Lehmann |
| Eunice Kennedy Shriver National Institute of Child Health and Human Development | K99HD088675 | Tatjana Trcek |

The funders had no role in study design, data collection and interpretation, or the decision to submit the work for publication.

## Author contributions
Kathryn E Kistler, Methodology, Investigation, Formal analysis, Validation, Writing-Original Draft; Tatjana Trcek, Conceptualization, Methodology, Investigation, Formal analysis, Validation, Software, Writing-Original Draft, Writing-Review&Editing, Funding Acquisition, Supervision; Thomas R Hurd, Conceptualization, Methodology, Investigation, Supervision; Ruoyu Chen, Methodology, Investigation, Formal analysis; Feng-Xia Liang, Methodology, Investigation; Joseph Sall, Masato Kato, Investigation; Ruth Lehmann, Conceptualization, Methodology, Writing-Review&Editing, Funding Acquisition, Resources, Supervision

## Author ORCIDs
Tatjana Trcek  https://orcid.org/0000-0003-4405-8733
Ruth Lehmann  http://orcid.org/0000-0002-8454-5651

## Decision letter and Author response
Decision letter https://doi.org/10.7554/eLife.37949.032
Author response https://doi.org/10.7554/eLife.37949.033

# Additional files

## Supplementary files
• Transparent reporting form
DOI: https://doi.org/10.7554/eLife.37949.030

All data generated or analysed during this study are included in the manuscript and supporting files. Source data has been provided for Figure 3 and Figure 3-figure supplement 1 (Supplementary file 1).

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
