## [Decision Letter]

Thank you for submitting your article "Phase transitioned nuclear Oskar promotes cell division of *Drosophila* primordial germ cells" for consideration by *eLife*. Your article has been reviewed by three peer reviewers, one of whom is a member of our Board of Reviewing Editors, and the evaluation has been overseen by Anna Akhmanova as the Senior Editor. The reviewers have opted to remain anonymous.

The reviewers have discussed the reviews with one another and the Reviewing Editor has drafted this decision to help you prepare a revised submission.

This manuscript focuses on understanding the formation and function of germ granules in developing *Drosophila* embryos, with a focus on the previously mysterious nuclear granules that are driven by Oskar protein. The authors use both embryos and an S2 cell-line to assess the properties of cytoplasmic and nuclear granules using standard assays such as sphericity, and recovery or loss from photobleaching to support that the granules behave as phase-separated condensates. Unlike the P granules of *C. elegans*, the germline granules of *Drosophila* appear to have at least two states – one that is more liquid-like and dynamic and one that is more gel-like and non-exchanging. A striking "hollow" structure is repeatedly seen even by TEM indicating some degree of structural organization akin to other systems (i.e. stress granules) and the shell may represent one of the distinct material states detected. The establishment of an S2 cell line-based assay is important for structure/function (with regard to granule formation) studies because mutants are defective in forming germline so it is difficult to assess mutant phenotypes and this should prove to be a useful system for future analysis of germline components. Using this assay, they characterize a variety of domains need redundantly for phase separation and identify and show that an NLS in Oskar is critical for the nuclear condensates and they go on and demonstrate this in animals using gene-edited mutants.

The most interesting result is the description of a possible function of the nuclear granules in regulating the division cycles of the germ cells. The authors show that the nuclear granules have distinct components and that they have different sizes from cytoplasmic granules but most interestingly, when the nuclear granules are eliminated with the NLS mutant, the division cycles of the GC become asynchronous, fewer cells divide and divisions take longer. This indicates that there is some function for the nuclear granules (and controls were done to support that this is not due to trivial impacts on protein stability or other known Oskar targets being misregulated) in coordinating cell cycle progression in this specific subpopulation of cells.

However, before publication, we would like you to address the following issues.

1) You mainly studied protein dynamics, and while the findings are suggestive of the claims regarding phase separated liquid-like and/or hydrogel-like character of germ granules, more evidence is required to justify the claims.

A) As further support for liquid-like character, it would be good if fusion of germ granules should be documented in *Drosophila* embryos (in addition to that demonstrated in cell culture lines).

B) Throughout the study, FRAP experiments have been conducted by photobleaching full germ granules. Additional FRAP experiments where only part of a germ granule has been photobleached should be done (possibly with large "Fused" germ granules, Figure 3). This will help test if the extent of immobile fraction is comparable in these two types of FRAP experiments (which probe rates of material exchange with the surroundings and that of internal rearrangements).

2) In line with this, you suggest that Oskar germ granules are more stable than other granules (subsection “Nuclear germ granules in S2R+ cells are phase transitioned condensates”, second paragraph). However, in Figure 4H, you report the FRAP *t_1/2_* and immobile fractions for a few P granule, p body, and stress granule components, some of which have similar or slower exchange rates and larger immobile fractions compared to Oskar. In fact 1 mammalian p body protein has 0% mobility and a large recovery *t_1/2_*. From the granule components currently noted in Figure 4H, it would actually suggest that many granules have a range of stable to dynamic components. You should also include other germ granules such as Buckyball which has been shown to have a slow and small mobile fraction (Boke, et al., 2016).(*side note, it is not clear how a protein with 0% mobility can have a meaningful *t_1/2_,* and this distorts the y axis of Figure 4H in a way that makes it look like granule components all have the same dynamics. It might be helpful to separate this data point from the others). (*they also cite Courchaine et al., 2016 for P granule FRAP; however it does not seem that these data were included in this paper).

3) With respect to your claim that the assembly of nuclear germ granules depends mainly on short-Oskar. But, germ granule-like punctate structures can be observed in the nucleus in absence of Oskar nuclear localization signal (see Figure 7B). You attribute this observation to "non-specific signal". It should be tested if some nuclear germ granules containing Vasa assemble in PGCs in absence of Oskar nuclear localization signal. It is worth noting that Vasa can form some nuclear puncta lacking Oskar (see Figure 3I). Some statistics describing the composition of nuclear germ granules in embryos and cell lines: Oskar (+) Vasa (+), Oskar (+) Vasa (-), or Oskar (-) Vasa (+) would be helpful.

One more point that was brought up was that we would be eager to see if cell cycle regulators are being sequestered in the nuclear granules as speculated and to us seeing this would make for a more impactful paper as it would point to a molecular function for condensates.

---

## [Author Response]

1) You mainly studied protein dynamics, and while the findings are suggestive of the claims regarding phase separated liquid-like and/or hydrogel-like character of germ granules, more evidence is required to justify the claims.A) As further support for liquid-like character, it would be good if fusion of germ granules should be documented in Drosophila embryos (in addition to that demonstrated in cell culture lines).

Using live imaging and laser scanning confocal microscopy we could not detect fusion of cytoplasmic germ granules or nuclear germ granules in the embryo (subsection “Cytoplasmic germ granules display properties of phase transitioned condensates”, last paragraph; subsection “Core germ granule proteins Oskar and Vasa form phase transitioned nuclear germ granules in primordial germ cells”, second paragraph). We did observe infrequent fusion of nuclear germ granule in S2R+ cells (Video 9; subsection “Nuclear germ granules in S2R+ cells are phase transitioned condensates”, first paragraph). Further, when fusion occurred it was slower than the fusion of characterized liquid-droplets such as P granules in *C. elegans* (Figure 4D, Video 8; see the aforementioned paragraph). We interpret this result as a behavior characteristic of phase transitioned droplets(Kato et al., 2012; Zhang et al., 2015) and in supports of our conclusion that cytoplasmic and nuclear germ granules in *Drosophila* are best be described as ‘phase transitioned condensates’. However, even when droplets phase transition, a fraction of the condensed protein can retain liquid-like properties and this population can be captured by FRAP. This has been shown for aged stress granules in mammalian cell lines (Lin et al., 2015; Niewidok et al., 2018; Wheeler et al., 2016), Whi3 granules in *Ashbya gossypii* (Zhang et al., 2015) and for hnRNPA2 granules (Xiang et al., 2015). To our knowledge, fusion of these specific granules has not been demonstrated in vivo despite the fact that they display (some) liquid-like properties. Thus, while informative, granule fusion is not the only or the ultimate indicator that granules behave like liquid droplets and exchange content with the environment. We used a number of different methods, FRAP, FLIP and photoconversion experiments to illustrate this point. These different types of measurements conclusively revealed a population (~30-40% ) of granular Oskar or Vasa with liquid properties (subsection “Miscellaneous phase separated condensates - distinct or the same?”, second paragraph). In the current version of the manuscript we complement these experiments by bleaching half of nuclear Oskar granules in S2R+ cells. These experiments revealed that Oskar protein can rearrange within the granule (Figure 4—figure supplement 1B; subsection “Nuclear germ granules in S2R+ cells are phase transitioned condensates”, second paragraph; Figure 4—figure supplement 1 legend; subsection “Fluorescence Recovery after Photobleaching (FRAP)”, second paragraph) further supporting our initial hypothesis that *Drosophila* germ granules display a liquid-like character.

Indeed, our findings are not unique, many granules that have been referred in the literature as ‘phase separated granules’ that form by LLPS display properties that appear liquid-like (mobile fraction recorded by FRAP) as well as hydrogel-like (immobile fraction recorded by FRAP) – see our new table that summarizes FRAP recordings for various proteins that populate such condensates (Figure 4—figure supplement 1C). As demonstrated in this table, granules differ in the percent of each of these fractions in the granule, and there are clear differences in the mobility of different proteins within the same type of granule. Recently, the McKnight group proposed that over time polymerization in liquid droplets increases, which leads to an increase of the hydrogel-like fraction while the rest of the granule protein retains more liquid-like properties(Xiang et al., 2015). In in vitro systems granule transitions can be observed over time, and reveal a decreased propensity of granules to fuse with age (Kato et al., 2012; Lin et al., 2015; Wheeler et al., 2016; Xiang et al., 2015; Zhang et al., 2015). We likely observe a similar behavior for Oskar granules; the liquid-like fraction is smaller than the hydrogel-like fraction and consequently, fusion of cytoplasmic or nuclear Oskar granules is rare. We have further discussed these issues in the revised Discussion section (subsection “Miscellaneous phase separated condensates - distinct or the same?”, second paragraph).

B) Throughout the study, FRAP experiments have been conducted by photobleaching full germ granules. Additional FRAP experiments where only part of a germ granule has been photobleached should be done (possibly with large "Fused" germ granules, Figure 3). This will help test if the extent of immobile fraction is comparable in these two types of FRAP experiments (which probe rates of material exchange with the surroundings and that of internal rearrangements).

We have performed these experiments and describe them in our revised manuscript (Figure 4—figure supplement 1B; subsection “Nuclear germ granules in S2R+ cells are phase transitioned condensates”, second paragraph; Figure 4—figure supplement 1 legend; subsection “Fluorescence Recovery after Photobleaching (FRAP)”, second paragraph). We focused on the fluorescence recovery of a partially bleached nuclear germ granule formed by mCherry:Osk in S2R+ cells as suggested by the reviewers. We recorded 28.6% higher mobile fraction than recorded when the entire granule was bleached (Figure 1C vs. Figure 4—figure supplement 1B). Thus, Oskar protein rearranges within the granule as well as with the granule environment, a result that further supports our observations that Oskar granules display liquid-like character. The kinetics of recovery (*t_1/2_*), which reports the average rate of protein exchange, is slower after partial bleaching (86.6 s vs. 10.2s) suggesting that this recovery is likely comprised of two rates, the rate of internal rearrangement which is likely a slow process and the rate of exchange of the protein with the granule environment which is fast (Figure 4E, F).

2) In line with this, you suggest that Oskar germ granules are more stable than other granules (subsection “Nuclear germ granules in S2R+ cells are phase transitioned condensates”, second paragraph). However, in Figure 4H, you report the FRAP t_1/2_ and immobile fractions for a few P granule, p body, and stress granule components, some of which have similar or slower exchange rates and larger immobile fractions compared to Oskar. In fact 1 mammalian p body protein has 0% mobility and a large recovery t_1/2_. From the granule components currently noted in Figure 4H, it would actually suggest that many granules have a range of stable to dynamic components. You should also include other germ granules such as Buckyball which has been shown to have a slow and small mobile fraction (Boke, et al., 2016).(*side note, it is not clear how a protein with 0% mobility can have a meaningful t_1/2_, and this distorts the y axis of Figure 4H in a way that makes it look like granule components all have the same dynamics. It might be helpful to separate this data point from the others). (*they also cite Courchaine et al., 2016 for P granule FRAP; however it does not seem that these data were included in this paper).

To more clearly present the data, we have removed the panel H from Figure 4 and instead summarized the data in a table format now found in the supplement (Figure 4—figure supplement 1C). First, we added the information about the exchange rates of Buck and Xvelo (Boke et al., 2016), as suggested by the reviewers. Where FRAP recordings were mentioned in the manuscript, we cite only the primary literature and not the reviews that referred to these recordings, as suggested by the reviewers (subsection “Nuclear germ granules in S2R+ cells are phase transitioned condensates”, second paragraph, Figure 4—figure supplement 1C legend).

Further, the table also includes FRAP recordings obtained for Vasa populating cytoplasmic and nuclear germ granules in the embryo (see also Figure 1—figure supplement 1C, Figure 2—figure supplement 1E).

Finally, in the original manuscript we stated that “given its large immobile fraction (~ 60%) Oskar cytoplasmic and nuclear germ granules in *Drosophila* and in S2R+ cells “are considerably more stable than other phase separated granules (Figure 4—figure supplement 1C)”. We have now removed the word “considerably” from the text to avoid over-statement.

The new table illustrates our conclusions that Oskar and Vasa in *Drosophila* germ granules have a smaller mobile fraction than reported for other the proteins that inhabit other membraneless granules (i.e.: PGL-1, GHL-1, PIE-1, TIA-1, PABP-1, LSm6, eIF4E, Dcp1a, Dcp1b; (Figure 4—figure supplement 1C)). The reviewers are correct in pointing out that most of these granules are also populated by proteins that seem to exchange less readily than Oskar or Vasa, however such proteins seem to be less abundant granule components. We did not intend to imply that *Drosophila* germ granules are the most stable condensates identified but rather to demonstrate that the two core functional components, Vasa and Oskar, which build these granules, on average appear less mobile than the proteins that populate other mebraneless granules. We have further clarified these observations in the manuscript (subsection “Miscellaneous phase separated condensates - distinct or the same?”).

3) With respect to your claim that the assembly of nuclear germ granules depends mainly on short-Oskar. But, germ granule-like punctate structures can be observed in the nucleus in absence of Oskar nuclear localization signal (see Figure 7B). You attribute this observation to "non-specific signal". It should be tested if some nuclear germ granules containing Vasa assemble in PGCs in absence of Oskar nuclear localization signal. It is worth noting that Vasa can form some nuclear puncta lacking Oskar (see Figure 3I). Some statistics describing the composition of nuclear germ granules in embryos and cell lines: Oskar (+) Vasa (+), Oskar (+) Vasa (-), or Oskar (-) Vasa (+) would be helpful.

To address the reviewers’ comments, we have collected embryos laid by mothers that do not express Oskar protein (*ΔOsk*) and immunostained them with an antibody against Oskar (Figure 7—figure supplement 1DL, subsection “Ablation of nuclear germ granules reduces cell divisions in PGCs”, first paragraph). This experiment supports our initial conclusion that the small speckles that appear in the PGCs of the *OskΔNLS* embryos are due to non-specific staining of the polyclonal (rabbit) Oskar antibody. Furthermore, we and others do not observe nuclear accumulation of Vasa in the absence of Oskar (Figure 3H, I; Figure 6E; Figure 5—figure supplement 1B (see *oskΔLOTUS* panel where the Vasa interaction domain was deleted in Oskar resulting in Vasa not to accumulate with Oskar in the nucleus). This result is also consistent with the observations from Anne Ephrussi’s group published in: (Jeske et al., 2017)); Figure 7—figure supplement 1C, E, F, subsection “Core germ granule proteins Oskar and Vasa form phase transitioned nuclear germ granules in primordial germ cells”, first paragraph; subsection “Expression of Short Oskar in cell lines reconstitutes nuclear germ granules”, fourth paragraph). Experiments from our and other labs have further shown that in the embryo, Oskar functions as the sole nucleator of granules. Oskar is not only necessary but also sufficient to initiate germ granule formation. In the absence of Oskar, Vasa does not form germ granules in the *Drosophila* embryo (Breitwieser et al., 1996; Ephrussi et al., 1991; Ephrussi and Lehmann, 1992; Jeske et al., 2015; Markussen et al., 1995). Consistent with these results is our observation that known *Drosophila* germ granule proteins did not form nuclear granules in the absence of Osk in S2R+ cells (Figure 3J, Figure 3—figure supplement 1G).

Finally, as reviewers suggested, we have statistically described the composition of nuclear germ granules in the embryo and in S2R+ cells (Figure 2—figure supplement 1A; Figure 3—figure supplement 1F; subsection “Core germ granule proteins Oskar and Vasa form phase transitioned nuclear germ granules in primordial germ cells”, first paragraph, subsection “Expression of Short Oskar in cell lines reconstitutes nuclear germ granules”, fourth paragraph). We have quantitatively characterized cytoplasmic germ granules previously(Trcek et al., 2015)). We find that Vasa:KuOr co-localized with Osk:GFP in 95.1% of nuclear germ granules (Figure 2—figure supplement 1A), in 90.3% cytoplasmic germ granules(Trcek et al., 2015) and in 80.3% nuclear germ granules in S2R+ cells (Figure 3—figure supplement 1F). In the embryo, all Vasa:KuOr nuclear granules also contained Osk:GFP (Figure 2—figure supplement 1A), indicating that Vasa can populate nuclear germ granules only when Oskar protein is present (Breitwieser et al., 1996; Lehmann, 2016). We did record that in S2R+ cells, 7.2% of Vasa positive granules appeared not to contain Oskar (Figure 3—figure supplement 1A). Given that Vasa cannot enter the nuclei or form germ granules by itself in S2R+ cells (see above) we prefer the hypothesis that those Vasa nuclear germ granules, which apparently lacked Oskar, were most likely populated by both proteins but that Oskar levels were not detected, rather than Vasa forming distinct granules devoid of Oskar protein (as discussed in the aforementioned paragraph).

One more point that was brought up was that we would be eager to see if cell cycle regulators are being sequestered in the nuclear granules as speculated and to us seeing this would make for a more impactful paper as it would point to a molecular function for condensates.

In our manuscript, we discuss that several known cell cycle regulators such as Mcm5, Mcm3, Dpa(Mcm4), Apc7 and Cdc16 were identified as specific interactors of Short Oskar by co-immunoprecipitation and are thus potential candidates for a function in the regulation of PGC division (Figure 3—figure supplement 1E). We agree with the reviewers that this is a very interesting and important aspect of our manuscript. Using fluorescent immunostaining we attempted to address whether these regulators co-localized with Oskar in nuclear germ granules as suggested by the reviewers. *Drosophila* specific antibodies were raised only against Mcm5 and DPA but had only be tested on Western blots and not in tissue(Su et al., 1997). For others, we acquired commercially available antibodies raised against mouse (Apc7, Cdc16) or human homologs (Mcm3, two variants). Unfortunately, none of the antibodies revealed a positive granule staining and these results remain inconclusive: the Mcm5 antibody never produced a nuclear stain although it is a nuclear protein at the end of G1 and S phase and Apc7, Cdc16 and Mcm3 might not be reactive against *Drosophila* homologs. Only DPA produced a nuclear stain but did not react with nuclear germ granules. Yet, even this result seems at best inconclusive without further experimentation. For example, we know from our experience with Vasa antibodies that these do not detect Vasa in nuclear germ granules, while we can clearly detect GFP or Kusabira Orange tagged Vasa in nuclear Oskar-granules in embryos and in tissue culture cells. Thus, our Vasa antibody somehow does not recognize Vasa in the nucleus. Since the same could be true for Dpa, we feel that at this stage the results are inconclusive. We have thus decided to not present and discuss these results in the manuscript.

We believe that addressing the question of how nuclear oskar may affect PGC division requires significant further investment in building tools and reagents and thus goes beyond the scope of our manuscript. We would like to stress that our manuscript already contains a large amount of new information by demonstrating that cytoplasmic and nuclear embryonic germ granules in *Drosophila* are phase transitioned condensates. Indeed, when one surveys the literature, only rarely is a clear function of phase transitions granules known. Indeed, even for the biophysically very well-studied *C. elegans* granules is a function in the embryo unclear as they are apparently not required for germ cell specification. In the case of *Drosophila*, embryonic germ granules have a clear and well-defined role: cytoplasmic granules function in germ cell specification and anterior-posterior patterning and our study now adds a new role for nuclear germ granules in early PGC cell division. Deciphering the mechanism and the regulator(s) by which these granules achieve this will be the work of future investigation.